# Single-cell sequencing of immune cells from anticitrullinated peptide antibody positive and negative rheumatoid arthritis

Xunyao Wu[1,2,11], Yi Liu [3,11], Shanzhao Jin [3,11], Min Wang[1,2,4,11], Yuhao Jiao[1], Bo Yang[5], Xin Lu[5], Xin Ji[1], Yunyun Fei[6], Huaxia Yang[6], Lidan Zhao[6], Hua Chen[6], Yaran Zhang[1], Hao Li[7], Peter E. Lipsky [8], George C. Tsokos [7✉], Fan Bai [3,9,10✉] & Xuan Zhang [4✉]

The presence or absence of anti-citrullinated peptide antibodies (ACPA) and associated disparities in patients with rheumatoid arthritis (RA) implies disease heterogeneity with unknown diverse immunopathological mechanisms. Here we profile CD45[+] hematopoietic cells from peripheral blood or synovial tissues from both ACPA+ and ACPA- RA patients by single-cell RNA sequencing and identify subsets of immune cells that contribute to the pathogenesis of RA subtypes. We find several synovial immune cell abnormalities, including up-regulation of *CCL13*, *CCL18* and *MMP3* in myeloid cell subsets of ACPA- RA compared with ACPA+ RA. Also evident is a lack of *HLA-DRB5* expression and lower expression of cytotoxic and exhaustion related genes in the synovial tissues of patients with ACPA- RA. Furthermore, the *HLA-DR15* haplotype (*DRB1/DRB5*) conveys an increased risk of developing active disease in ACPA+ RA in a large cohort of patients with treatment-naive RA. Immunohistochemical staining shows increased infiltration of CCL13 and CCL18-expressing immune cells in synovial tissues of ACPA- RA. Collectively, our data provide evidence of the differential involvement of cellular and molecular pathways involved in the pathogenesis of seropositive and seronegative RA subtypes and reveal the importance of precision therapy based on ACPA status.

[1] Clinical Immunology Center, State Key Laboratory of Complex Severe and Rare Diseases, Peking Union Medical College Hospital, Chinese Academy of Medical Sciences and Peking Union Medical College, Beijing, China. [2] Department of Medical Research Center, Peking Union Medical College Hospital, Chinese Academy of Medical Sciences and Peking Union Medical College, Beijing, China. [3] Biomedical Pioneering Innovation Center (BIOPIC), School of Life Sciences, Peking University, Beijing, China. [4] Department of Rheumatology, Beijing Hospital, National Center of Gerontology; Institute of Geriatric Medicine, Chinese Academy of Medical Sciences, Beijing, China. [5] Department of Orthopaedics, Peking Union Medical College Hospital, Chinese Academy of Medical Sciences and Peking Union Medical College, Beijing, China. [6] Department of Rheumatology and Clinical Immunology, Peking Union Medical College Hospital, Chinese Academy of Medical Sciences and Peking Union Medical College, The Ministry of Education Key Laboratory, Beijing, China. [7] Division of Rheumatology, Beth Israel Deaconess Medical Center, Harvard Medical School, Boston, MA, USA. [8] RILITE Research Institute and AMPEL BioSolutions, Charlottesville, VA, USA. [9] Center for Translational Cancer Research, First Hospital, Peking University, Beijing, China. [10] Beijing Advanced Innovation Center for Genomics (ICG), Peking University, Beijing, China. [11]These authors contributed equally: Xunyao Wu, Yi Liu, Shanzhao Jin, Min Wang. ✉email: gtsokos@bidmc.harvard.edu; fbai@pku.edu.cn; zxpumch2003@sina.com

Characterized by joint inflammation and bone destruction, rheumatoid arthritis (RA) is considered a heterogeneous disease, which can be categorized by the presence or absence of anticitrullinated-peptide antibodies (ACPA)[1]. ACPAs are disease-specific biomarkers for RA and are present in approximately two-thirds of these patients[2]. Genetic, clinical, and therapeutic differences exist between ACPA- and ACPA+ RA patients[3–5]. However, the differences in immunopathogenesis between these two RA subtypes are not well understood. Citrullinated autoantigens in ACPA+ RA probably activate monocyte/macrophages directly by forming immune complexes (IC) with ACPA. ICs subsequently trigger a proinflammatory cytokine response in macrophage colony-stimulating factor (M-CSF) differentiated macrophages and induce TNF secretion by blood and synovium-derived macrophages in patients with RA[6,7]. Extracellular citrullinated proteins presented by antigen-presenting cells (APCs) to auto-reactive T and B cells trigger an adaptive immune response[8]. The study of peripheral blood-derived citrullinated peptide-specific B cells in RA has revealed that these cells have a memory phenotype, expressing IgA or IgG, and are present at relatively high frequencies[9]. The synovial T cell repertoire differs between ACPA+ and ACPA- RA[10]. Furthermore, ACPA might contribute to bone loss and the development of joint pathology by facilitating synovium-derived fibroblast migration[11]. As the immunopathogenesis of ACPA- RA is still not clear, an understanding of the mechanisms involved is warranted to enable the management of this disease subtype. The single-cell technologies have helped to identify inflammatory cell states and specific immune cell populations associated with RA[12,13].

Here we apply single-cell RNA sequencing (scRNA-seq) to characterize the cell composition, proportion, gene expression signature, and developmental trajectories of CD45+ cells in the peripheral blood and synovial membrane of ACPA+ and ACPA- RA patients. We present this single-cell immunological landscape and identify immune cell types in ACPA- RA that might contribute to disease development.

## Results

### The immunological landscape of immune cells from ACPA+ and ACPA- RA.
To investigate the heterogeneity of immune cell phenotype in different subtypes of RA patients, we carried out scRNA-seq on CD45+ peripheral blood mononuclear cells (PBMCs) isolated from healthy controls (HCs) ($n = 4$), PBMCs and synovial tissue mononuclear cells (STMCs) isolated from ACPA+ ($n = 10$), and ACPA- ($n = 10$) RA patients (Fig. 1a). The studied patients were not taking disease-modifying antirheumatic drugs (DMARDS), corticosteroids, or targeted therapies at the time of sampling (some patients chose to use physical therapies, such as heat incubation or acupuncture to alleviate pain). After data preprocessing and quality control, we obtained single-cell transcriptomes of 135,429 immune cells from PBMCs and 71,073 immune cells from STMCs. Using unsupervised graph-based clustering, we identified 25 clusters in PBMCs, which consisted of three major populations, B cells, T/Natural Killer (NK) cells, and myeloid cells (monocytes and dendritic cells), by examining the expression of canonical markers CD3D, CD79A, CD14, FCER1A, FCGR3A, IL3RA, NKG7, PPBP, and MZB1 (Fig. 1b, Supplementary Fig. 1a).

We identified a total of four clusters of B cells in PBMCs from all individuals by comparing the expression of IGHD, CD27, IGHM, IGHG1, IGHG3, IGHA1, and MZB1; these B cell clusters included naïve B cells, memory-unswitched B cells (mem-unsw B), memory-switched B cells (mem-sw B), and plasma B cells (Supplementary Fig. 1b). We found that the proportion of B cell

populations were similar in RA patients in comparison to those in HCs (Fig. 1c). We identified a total of eight distinct subsets of T/NK cells in PBMCs. Among T cells, naïve T cells expressed CCR7 and SELL, exhausted T cells expressed high amounts of PDCD1, LAG3, and TIGIT, (Interferon) IFN-act T cells expressed IFI6, ISG15, and STAT1, GZMK[hi] CD4 T cells and GZMK[hi] CD8 T cells highly expressed GZMK, and cytotoxic T cells expressed cytotoxic genes (NKG7, PRF1, GNLY, and GZMB) (Supplementary Fig. 1b). We observed lower proportions of naïve T and higher proportions of cytotoxic T cells in ACPA+ RA compared with HCs. No significant differences were found in the proportions of T cell subsets between ACPA- and ACPA+ RA (Fig. 1c, Supplementary Fig. 1c). Analysis of myeloid cells in the PBMC on the basis of differential expression of CD14, FCGR3A, FCER1A, IRF8, IL3RA, HLA-DRB5, S100B, and IFN activation markers (ISG15, IFI6, IFI44, and STAT1) revealed no significant differences in Monocyte/Dendritic cells (Mono_DC) populations in ACPA- RA compared with ACPA+ RA and HCs, respectively (Fig. 1c, Supplementary Fig. 1b).

A total of 20 cell clusters were identified in STMCs and four major populations, including B cells, T/NK cells, dendritic cells (DCs), and macrophages were defined by examining the expression of CD79A, CD3D, NKG7, C1QA, CD1C, IL3RA (Fig. 1d, Supplementary Fig. 2a). Two B cell subsets were identified: Memory B (GPR183) and Plasma B (MZB1, JCHAIN); six T/NK cell subsets were identified: central memory T (TCM, CCR7, GRP183), effector memory T (TEM, GPR183, IL7R), GZMK[hi] T (GZMK), cytotoxic T (CD3E, GNLY, NKG7, GZMB), exhausted T (PDCD1, TIGIT, LAG3), and NK (NKG7, GNLY, GZMB); four DC subsets were identified: cDC (CD1C, FCER1A), S100B[hi] DC (S100B), LAMP3+ DC (LAMP3) and pDC (IL3RA, IRF8 and GZMB); Seven macrophage subsets were identified: CCL+ macrophages (CCL13, CCL3, CCL18), CCL3+ CCL3L3+ macrophages (CCL3L3, CCL13, CCL3, and CCL18), IFN-act macrophages (ISG15, IFI6, IFI44, and STAT1), M1 macrophages (IL1B), IL1B+ S100A12+ macrophages (IL1B, S100A12), MT1G+ macrophages (MT1G) and SPP1[hi] macrophages (SPP1) (Supplementary Fig. 2b). Analysis of subpopulations of B cells, T cells, DCs, and macrophages revealed lower proportions of NK, exhausted T, SPP1[hi] macrophages, and relatively higher proportions of pDC in the synovial membrane of ACPA- compared with ACPA+ RA patients (Fig. 1e, Supplementary Fig. 2c). In summary, we observed a lower proportion of cytotoxic cells in PBMC and synovial membrane of ACPA- RA patients compared with ACPA+ RA patients.

### ACPA- RA lacks B cell expression of HLA-DRB5.
To characterize the specific signature of immune/inflammatory cells in the two RA subtypes, we examined the cellular-subtype proportions within each cell populations from PBMCs and synovial tissues (ST). Using graph-based clustering, we sub-grouped PBMC-derived B cells into seven cell subtypes and designated each subset according to their "core gene signature." Based on IGHD, CD27, IGHM, IGHG1, IGHA1, GNLY, GPR183, IGHG3, IGHG4, MZB1, and HLA-DRB5 expression, we defined naïve B, Mem-unsw B, Mem-sw B, Plasma B, IGHG4+ Plasma B, HLA-DRB5+ MZB1- B, and HLA-DRB5+ Plasma B subsets (Fig. 2a-b), respectively. Next, we compared the proportions of B cell subsets in different RA subtypes. It was noteworthy that HLA-DRB5+ Plasma B cells, expressing HLA-DRB5, IGHM, IGHG1, and MZB1, were significantly decreased; however, IGHG4+ Plasma B cells, distinctly expressing IGHG4 from other plasma B subsets, were significantly increased in the peripheral blood of ACPA- RA patients (Fig. 2c, Supplementary Fig. 3a). We sub-grouped ST-derived B cells into eight-cell subtypes (Fig. 2d-e). We identified

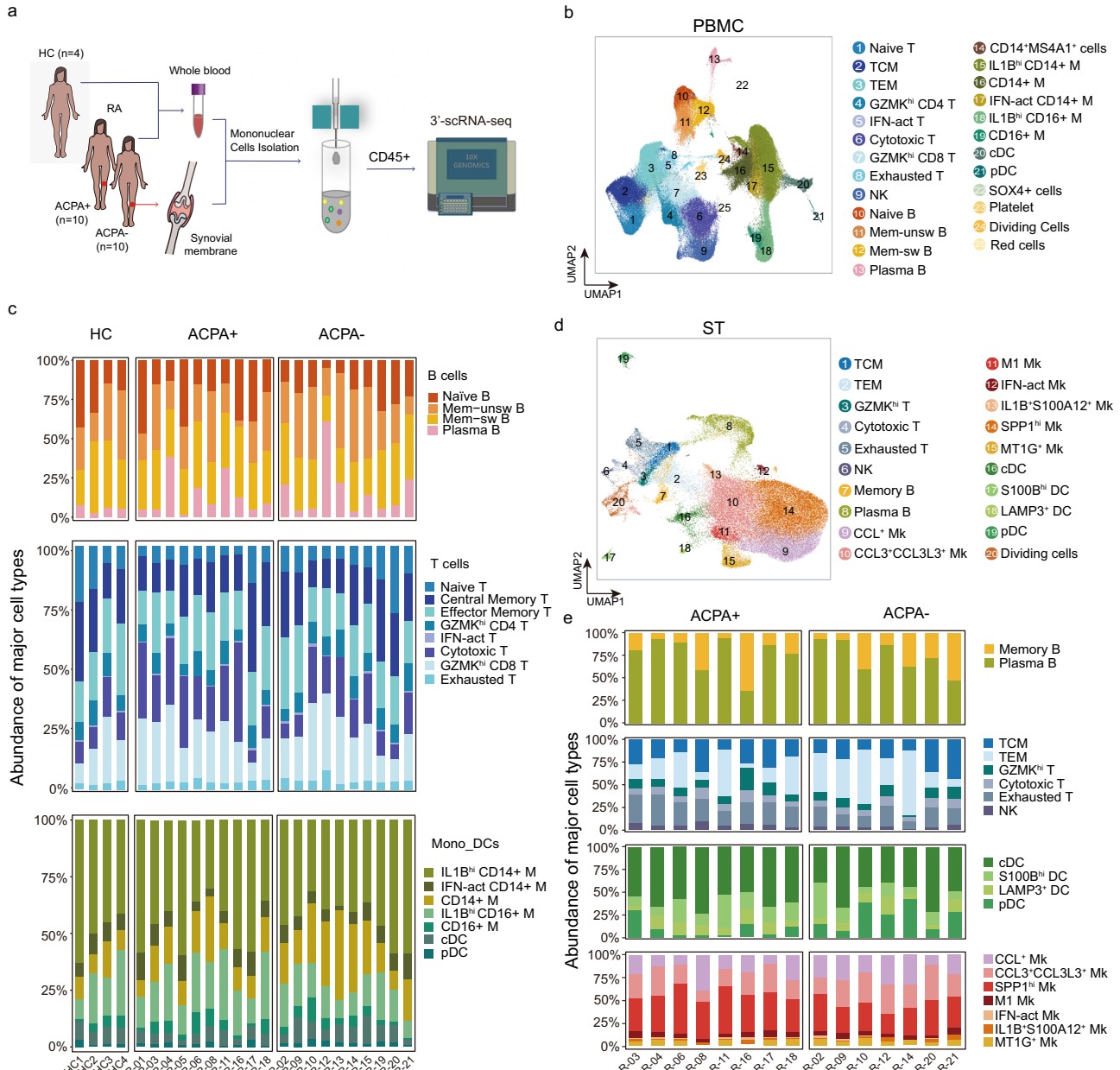

**Fig. 1 Clustering and classification of immune cells in peripheral blood and synovium from patients with RA. a** Overview of the experimental workflow. HC: Healthy Control; ACPA: anti-citrullinated-peptide antibodies; scRNA-seq: single-cell RNA sequencing. **b** UMAP visualization of pooled scRNA-seq data of 135,429 CD45$^+$ cells from PBMC (Peripheral blood mononuclear cells). We identified 25 clusters, including B cells (four clusters), T/NK cells (nine clusters), and myeloid cells (seven clusters). **c** Relative percentage of cell subtypes in PBMC across donors, grouped by ACPA type. **d** UMAP visualization of pooled scRNA-seq data of 71,073 CD45$^+$ cells from STMC (synovial tissue mononuclear cells). We identified 20 clusters, including B cells (two clusters), T cells (6 clusters), dendritic cells (four clusters), and macrophages (seven clusters). **e** Relative percentage of cell subtypes in STMC across donors, grouped by ACPA type.

three memory B and five plasma cell subtypes in ST, respectively. Among them, *HLA-DRB5*$^+$ and *HLA-DRB5*$^+$ *HOPX*$^+$ memory B cells, which were absent in more than half (four of seven) of STs from ACPA- RA patients, shared common highly expressed genes (*GPR183* and *HLA-DRB5*) and distinct *HOPX* expression in *HLA-DRB5*$^+$ *HOPX*$^+$ memory B cells (Fig. 2d–f, Supplementary Fig. 3b). *CCL*$^+$ memory B cells, which express a set of chemokine genes that include *CCL3*, *CCL13*, and *CCL18*, were enriched in the synovial tissues (Fig. 2d–e). Relatively higher proportions of *CCL*$^+$ memory B cells were observed in the synovial membrane of ACPA- RA patients but this difference was not statistically significant from that in ACPA+ RA patients (Fig. 2f).

To further explore the continuum of class-switching states in B cell subsets, we carried out pseudotime ordering of single cells using Monocle 3. Starting from naïve B cells, we reconstructed a trajectory with three branches in PBMCs. We noted that although *HLA-DRB5*$^+$ Plasma B subsets expressed plasma B-related genes (*MZB1*), they are enriched for "peptide antigen binding" and "antigen processing and presentation," which resemble an "intermediate" subset (Fig. 3a). In HCs, progression proceeded from naïve B cells to either *HLA-DRB5*$^+$ Plasma B cells and then to plasma cells or along the trajectory from Mem-unsw B to Mem-sw B (Fig. 3b). In contrast, in PBMCs of RA patients, we identified an alternative pathway from naïve B cells to *IGHG4*$^+$

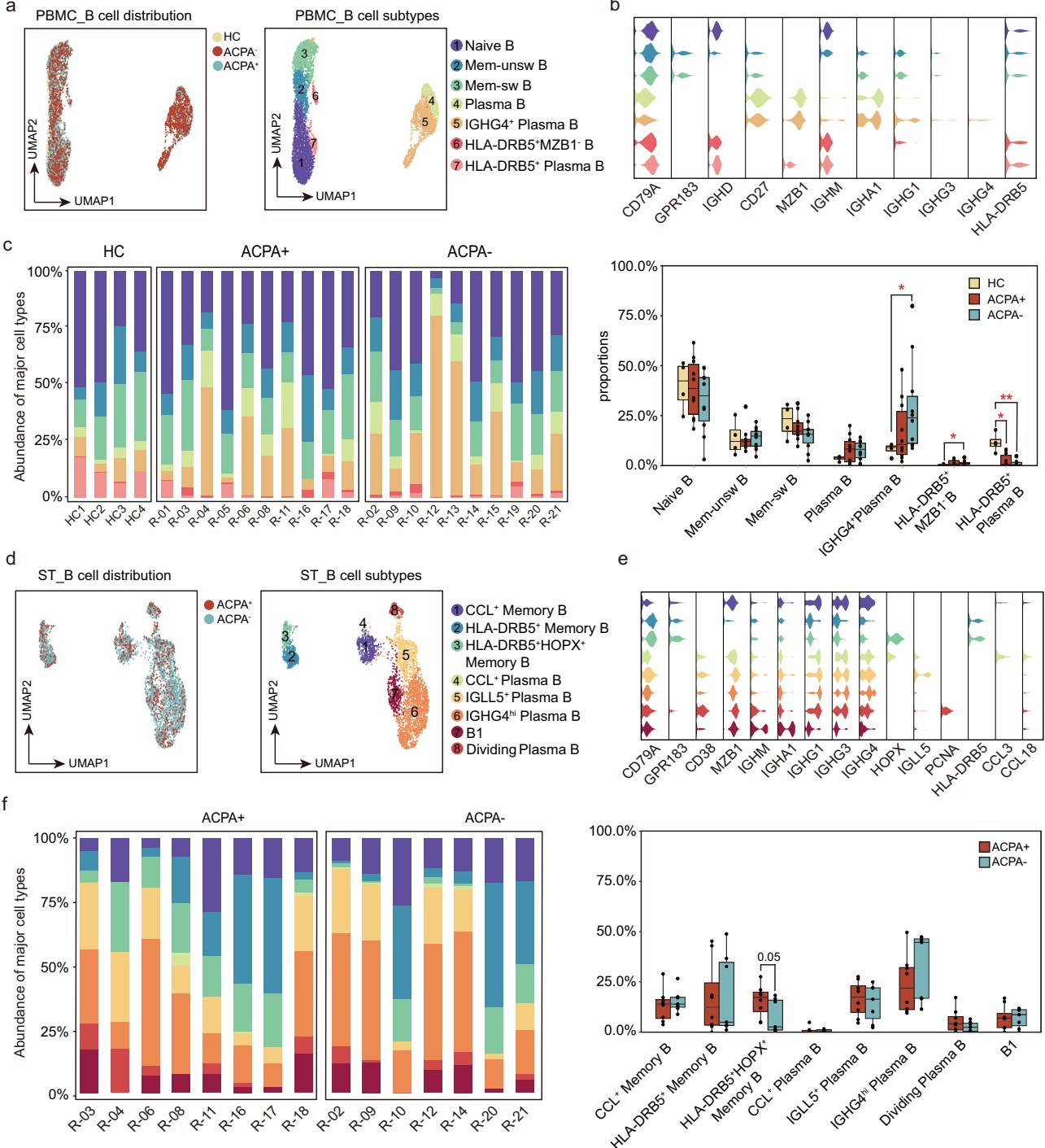

**Fig. 2 Lack of HLA-DRB5 expression by B cells from patients with ACPA- RA. a** UMAP visualization of B cell subsets from PBMC. We identified 7 B cell clusters across 6152 cells. Cells are marked by ACPA type (left) and cell subtype (right). **b** Violin plots showing marker genes across PBMC B cell subtypes. The $y$ axis represents log-scaled normalized counts. **c** Left: bar plots showing the relative percentage of PBMC B cell subtypes for each sample as in a. Right: box plots showing the proportions of each PBMC B cell subtypes across ACPA groups. Cell types showed enrichment in ACPA+ or ACPA- subgroups are marked with *$P$ values were calculated by the two-sided Wilcoxon test. *$p < 0.05$ (from left to right, $p = 0.024, 0.036, 0.013$), **$p < 0.01$ ($p = 0.0058$). $n = 4$ for HC, $n = 10$ for ACPA+ group, and $n = 10$ for ACPA- group. **d** UMAP visualization of B cell subsets from STMC. We identified 8 B cell clusters across 4151 cells. Cells are marked by ACPA type (left) and cell subtype (right). **e** Violin plots showing marker genes across STMC B cell subtypes. The $y$ axis represents log-scaled normalized counts. **f** Left: bar plots showing the relative percentage of STMC B cell subtypes for each sample as in **d**. Right: box plots showing the proportions of STMC each B cell subtypes across ACPA groups. Cell types showed enrichment in ACPA+ or ACPA- subgroups are marked. $P$ values were calculated by the two-sided Wilcoxon test. $n = 10$ for ACPA+ group, and $n = 10$ for ACPA- group.

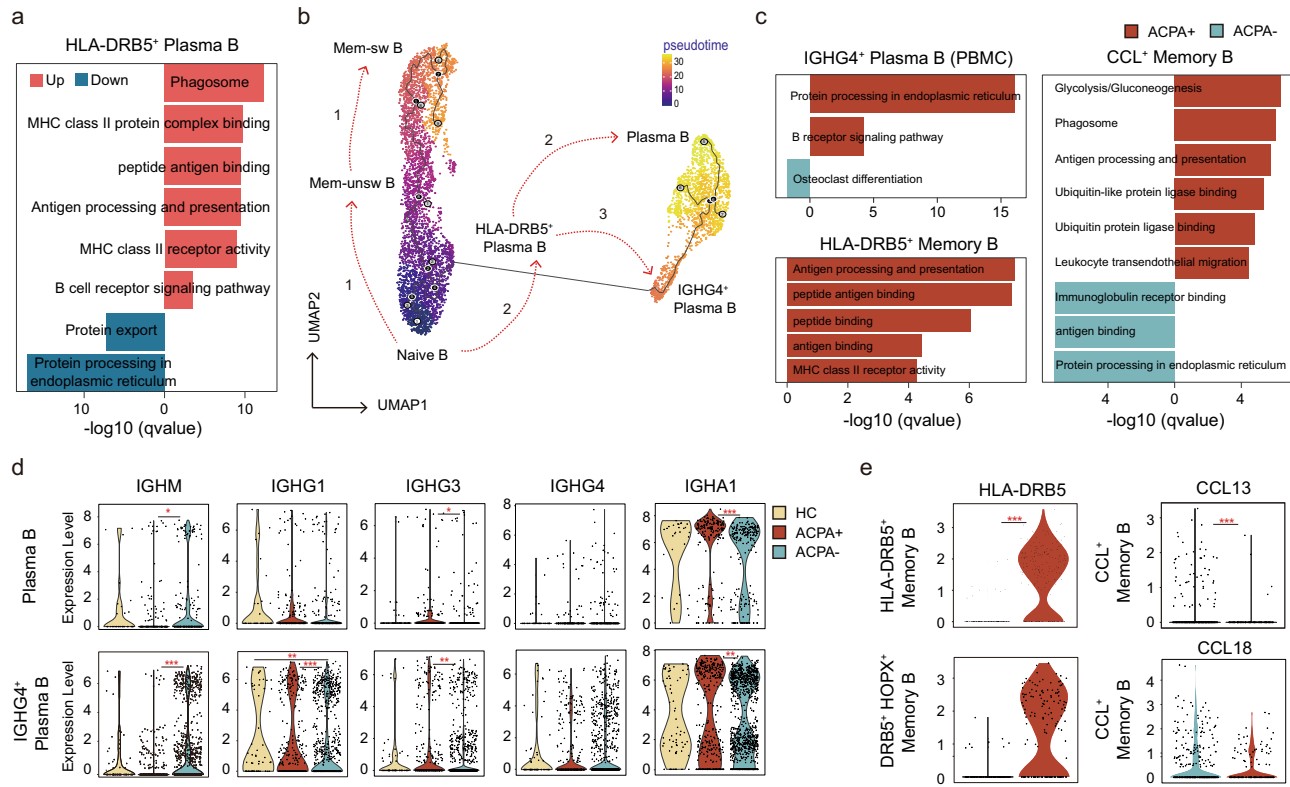

**Fig. 3 Abnormal B cell class-switching in patients with ACPA- RA. a** Bar plots showing the DEG (differential expressed genes) enriched Kyoto Encyclopedia of Genes and Genomes (KEGG) pathways and Gene Ontology (GO) terms of HLA-DRB5+ Plasma B. *P* values were calculated using Hyper Geometric Test, and q values were adjusted from p values using the Benjamini-Hochberg method. **b** Pseudotime reconstruction and developmental trajectory of B cells in PBMCs inferred by Monocle 3. Predicted developmental trajectories are shown using arrows. **c** Bar plots showing the DEG (differential expressed genes) enriched Kyoto Encyclopedia of Genes and Genomes (KEGG) pathways and Gene Ontology (GO) terms of specific B cell subtypes. *P* values were calculated using Hyper Geometric Test, and q values were adjusted from p values using the Benjamini-Hochberg method. **d** Violin plots showing the differences in immunoglobin gene expression between ACPA- and ACPA+ B cells in PBMC. The y axis represents log-scaled normalized counts. *P* values were calculated by the two-sided Wilcoxon test. *p < 0.05, **p < 0.01, ***p < 0.001. **e** Violin plots showing the differences in *HLA-DRB5* gene and chemokine gene expression between ACPA- and ACPA+ B cells in STMC. Asterisks indicate the significance. The y axis represents log-scaled normalized counts. *P* values were calculated by the two-sided Wilcoxon test. *p < 0.05, **p < 0.01, ***p < 0.001.

Plasma B cells along with the two branches observed in HCs (Fig. 3b).

We next compared the functional and molecular differences of *IGHG4*+ Plasma B, *HLA-DRB5*+ Memory B, and CCL+ Memory B between ACPA- and ACPA+ RA patients. We found a lower transcriptional profile, with enrichment for "protein processing in endoplasmic reticulum" and "B receptor signaling pathway" but a relative higher transcriptional profile, with enrichment for "osteoclast differentiation" of *IGHG4*+ Plasma B in ACPA- RA patients compared with ACPA+ RA patients (Fig. 3c). ACPA- RA patients-derived plasma B cells in peripheral blood displayed a higher expression of *IGHM* and lower expressions of *IGHG3* and *IGHA1* (Fig. 3d). *HLA-DRB5*+ memory B cells from ACPA- RA patients were absent for *HLA-DRB5* expression and exhibited a lower transcriptional profile, with enrichment for "antigen processing and presentation", "peptide antigen binding", "peptide binding", "antigen binding", and "MHC class II receptor activity" compared with ACPA+ RA patients (Fig. 3c, e). CCL+ memory B cells from ACPA- RA patients were additionally transcriptionally distinct, being enriched for "immunoglobulin receptor binding", "antigen binding", and "protein processing in endoplasmic reticulum" but lower for "glycolysis/gluconeogenesis", "phago-some", and "antigen processing and presentation activity" (Fig. 3c). In summary, we demonstrated a lack of an "intermediate" *HLA-DRB5*+ Plasma B and abnormal *IGHG4*+ Plasma B in peripheral blood and a lack of *HLA-DRB5* expression

as well as impaired "antigen processing and presentation activity" on memory B cells in ST of ACPA- RA.

**High level of *CCL13*, *CCL18*, and *MMP3* expression by myeloid cells in ACPA- RA ST**. Based on *FCER1A*, *CD1C*, *IL3RA*, *GZMB*, *HLA-DRB5*, *CD14*, *CLEC9A*, and *IRF8* expression, we identified seven subtypes of dendritic cells (DC) in PBMCs (Supplementary Fig. 4a–b). A significantly higher proportion of *HLA-DRB5*- plasmacytoid dendritic cells (pDC), *HLA-DRB5*- DC, and rela-tively lower proportion of *HLA-DRB5*+ pDC was found in ACPA- RA patients (Supplementary Fig. 4c, d). We further identified eight subtypes of dendritic cells in ST, including DC_macrophage (*FCER1A*, *CD1C*, and *CD14*), cDC (*FCER1A*, *CD1C*), *CLEC9A*+ DC (*CLEC9A*, *S100B*), *IL3RA*+ DC (*FCER1A*, *IL3RA*), *CXCL10*+ *LAMP3*+ DC (*CD1C*, *LAMP3*, and *CXCL10*), *LAMP3*+ DC (*LAMP3*), pDC (*IL3RA*, *GZMB*), CCL+ pDC (*IL3RA*, *GZMB*, *CCL3*, *CCL13*, and *CCL18*) (Fig. 4a, b, Supple-mentary Fig. 5a, b). No obvious differences in the proportions of DC subsets in ST were observed between ACPA- and ACPA+ RA (Fig. 4c). Notably, *CCL13*, *CCL18*, and *MMP3* were the most significantly upregulated genes, and *HLA-DRB5*, *SPP1*, *BRI3* expression were absent in ACPA- RA ST-derived DC subsets (Fig. 4d). When we further applied Kyoto Encyclopedia of Genes and Genomes (KEGG) pathway and Gene Ontology (GO) enrichment analyses, we found "phagosome", "glycolysis/

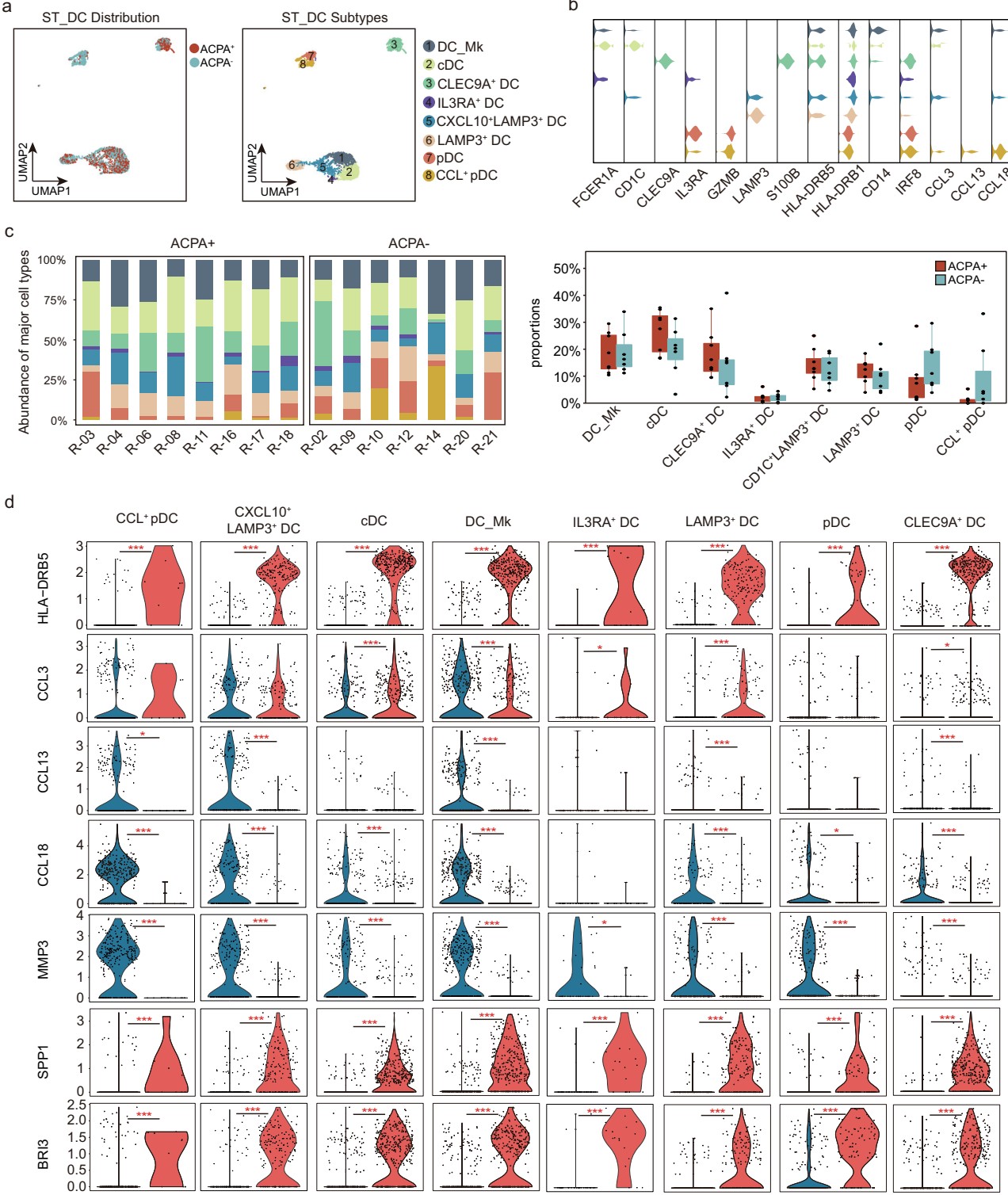

**Fig. 4 Enrichment of CCL+ DC subsets in synovial tissue from patients with ACPA- RA. a** UMAP visualization of dendritic cell (DC) subtypes from the synovium. We identified eight DC clusters across 3369 cells. Cells are marked by ACPA type (left) and cell subtypes (right). **b** Violin plots showing marker genes across STMC DC subtypes. The *y* axis represents log-scaled normalized counts. **c** Left: bar plots showing the relative percentage of STMC DC subtypes for each sample as in **a** Right: box plots showing the proportions of each STMC DC subtypes across ACPA groups. *n* = 10 for ACPA+ group, and *n* = 10 for ACPA- group. **d** Violin plots showing the differences in interested gene expression between ACPA- and ACPA+ DC subtypes in STMC. Asterisks indicate the significance. The *y* axis represents log-scaled normalized counts. *P* values were calculated by the two-sided Wilcoxon test. *p < 0.05, **p < 0.01, ***p < 0.001.

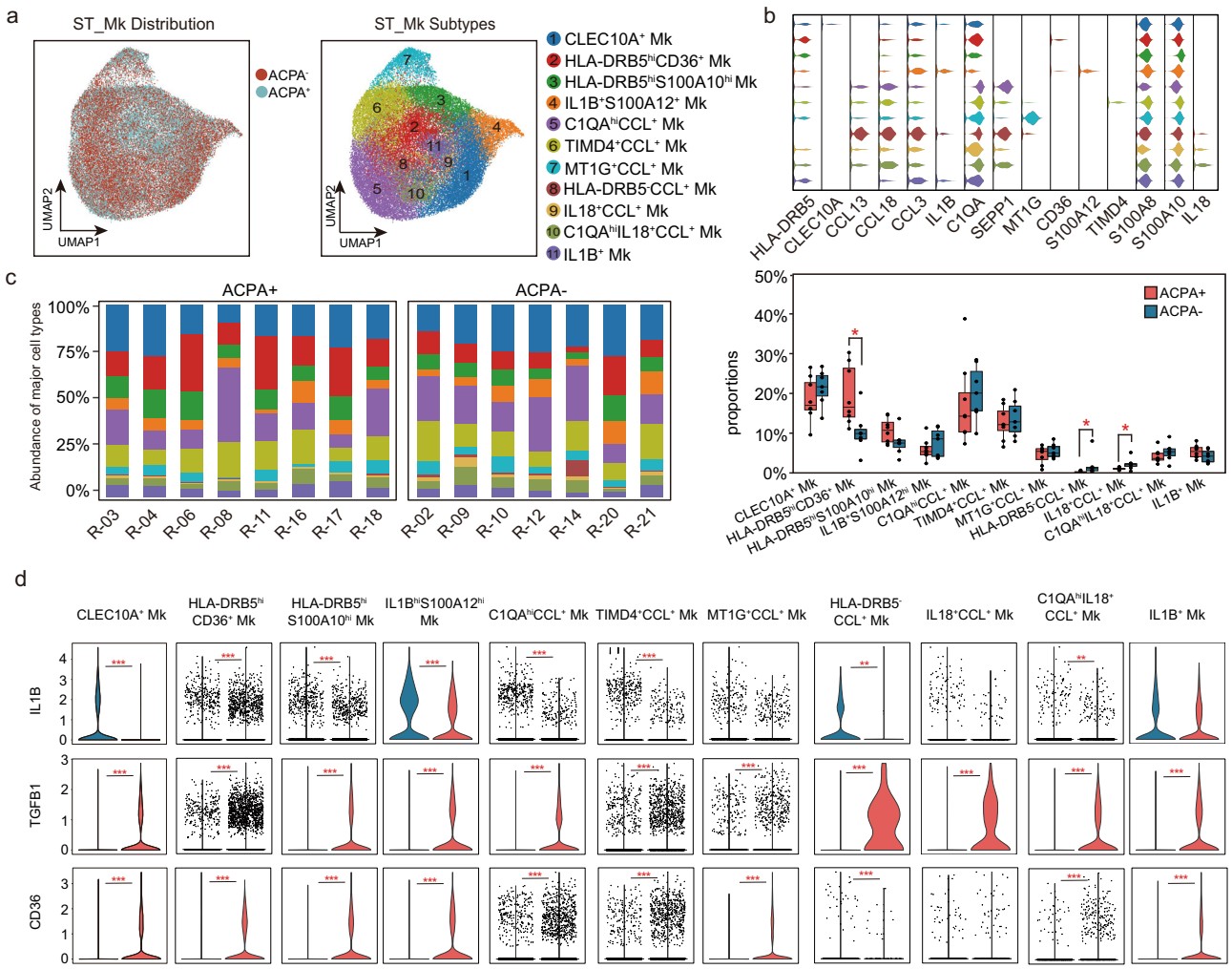

**Fig. 5 Identification of macrophage subsets in the synovium of patients with RA. a** UMAP visualization of macrophages from the synovium. Among 46,973 cells, we identified 11 macrophage subtypes. Cells are marked by ACPA type(left) and cell subtypes (right). **b** Violin plots showing marker genes across cell subtypes in A. The y axis represents log-scaled normalized counts. **c** Left: Bar plots showing the relative percentage of STMC macrophage subtypes for each sample as in a. Right: Box plots showing the proportions of each STMC macrophage subtypes across ACPA groups. Cell types showed enrichment in ACPA+ or ACPA- subgroups are marked with *P values were calculated by the two-sided Wilcoxon test. *p < 0.05 (from left to right, p = 0.021, 0.043), **p < 0.01 (p = 0.0093). n = 10 for ACPA+ group, and n = 10 for ACPA- group. **d** Violin plots showing the differences in the expression of IL1B, TGFB1, and CD36 between ACPA- and ACPA+ macrophage subtypes in STMC. Asterisks indicate the significance. P values were calculated by the two-sided Wilcoxon test. *p < 0.05, **p < 0.01, ***p < 0.001.

gluconeogenesis", "antigen processing and presentation" were mainly downregulated in the ST DC subsets of ACPA- RA compared with ACPA+ RA (Supplementary Fig. 5c). We identified eight monocyte clusters in PBMCs based on canonical marker expression (Supplementary Fig. 6a–b). CD14 monocytes, IFN-act CD14 monocytes, *S100A8*[hi]*S100A12*[hi] CD14 monocytes, *HLA-DRB5*[+] CD14 moncytes, and CD14[hi]CD16[low] monocytes all comprise CD14 monocytes and display high expression of *CD14* genes as well as upregulation of *S100A8*, *S100A12*, *HLA-DRB5*, and IFN-act-related genes (*ISG15*, *IFI6*, *IFI44*, and *STAT1*) or *FCGR3A* by differential gene analysis (Supplementary Fig. 6a, b). IFN-act CD16[+] monocytes and CD14[dim]CD16[high] monocytes are part of the CD16[+] monocyte population; they share expression of *FCGR3A*, but higher IFN-act-related genes or CD14 expression by differential gene analysis (Supplementary Fig. 6a, b). We did not observe an obvious enrichment of distinct monocyte populations in the peripheral blood of ACPA- RA patients (Supplementary Fig. 6c, d). Clustering analysis identified 11 macrophage subtypes visualized using UMAP (Fig. 5a, b). *CLEC10A*[+] macrophages were characterized by distinct expression of *CLEC10A*

(Fig. 5b). *IL1B*[+] *S100A12*[+] macrophages and *IL1B*[+] macrophages shared common highly expressed *IL1B* gene and distinct *S100A12* expression by *IL1B*[+] *S100A12*[+] macrophages (Fig. 5b). Another six macrophage subtypes resident in the synovial membrane shared common expression of the CCL chemokine genes (*CCL3*, *CCL13*, and *CCL18*) were characterized by their distinct expression of *C1QA*, *TIMD4*, *MT1G*, and *IL18*. We defined these macrophage populations as *C1QA*[hi]CCL[+] macrophages, *TIMD4*[+] CCL[+] macrophages, *MT1G*[+] CCL[+] macrophages, *HLA-DRB5*[-]CCL[+] macrophages, *IL18*[+] CCL[+] macrophages, and *C1QA*[hi]*IL18*[+]CCL[+] macrophages (Fig. 5a-b). We further identified two subtypes of *HLA-DRB5*[hi] macrophage and named them *HLA-DRB5*[hi]*CD36*[+] macrophages and *HLA-DRB5*[hi]*S100A10*[hi] macrophages, respectively (Fig. 5a–b). The proportion of *HLA-DRB5*[hi]*CD36*[+] macrophages was significantly lower in ST of ACPA- RA (Fig. 5c). Of CCL[+] macrophage subsets, the proportions of *HLA-DRB5*[-]CCL[+] macrophages and *IL18*[+] CCL[+] macrophages were higher in the synovial membrane of ACPA- RA (Fig. 5c). Moreover, we found that ACPA- RA ST macrophage subsets displayed proinflammatory (M1 macrophage)

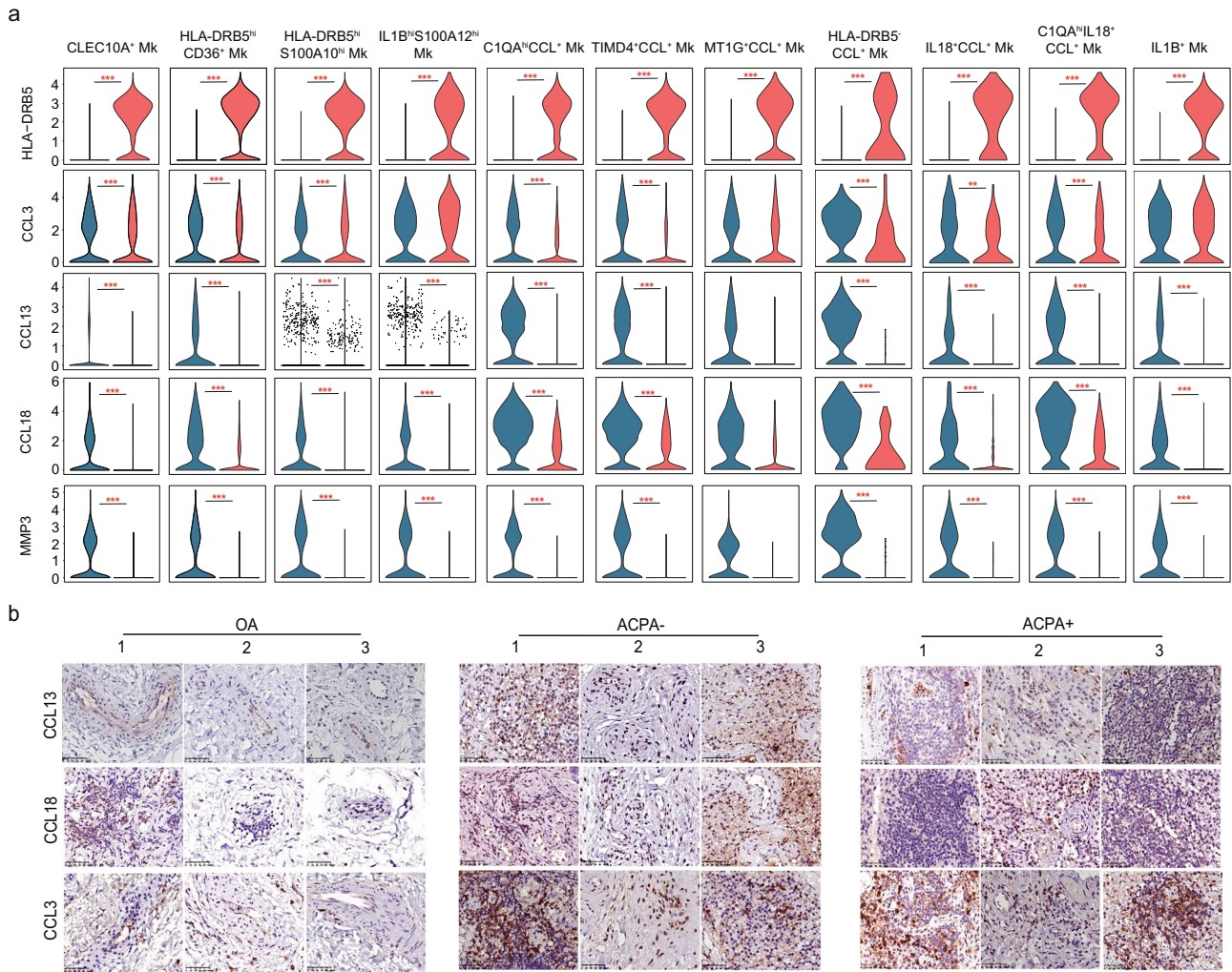

**Fig. 6 CCL⁺ macrophages are enriched in the synovial tissue of patients with ACPA- RA. a** Violin plots showing the differences in the expression of HLA-DRB5, CCL3, CCL13, CCL18, and MMP3 between ACPA- and ACPA+ macrophage subtypes in STMC. Asterisks indicate the significance. *P* values were calculated by the two-sided Wilcoxon test. *$p < 0.05$, **$p < 0.01$, ***$p < 0.001$. **b** Immunohistochemistry of CCL3, CCL13, and CCL18 expression in the synovial tissue of 3 OA, 3 ACPA- RA, and 3 ACPA+ RA patients. The immunohistochemistry results were carried out in one independent experiment of three biological replicates from each group.

features with upregulated *IL1B* gene expression, and almost absent *TGFB1* gene expression (Fig. 5d). Also, the lower expression of *CD36* in all macrophage subsets indicated the impaired phagocytic activity in ACPA- RA compared with ACPA+ RA (Fig. 5d). In accordance with ACPA- RA ST DC subsets, *CCL13*, *CCL18*, and *MMP3* were also the most significantly upregulated genes and *HLA-DRB5* was absent in ACPA- RA ST macrophage subsets (Fig. 6a). We performed immunohistochemical staining to explore the expression patterns of chemokines (CCL13, CCL18, and CCL3) in the synovial membrane of RA subgroups, while the osteoarthritis (OA) synovial membrane was used as controls. We observed increased infiltration of CCL13 and CCL18-expressing immune cells in the synovial membrane of RA patients, especially in ACPA- RA (Fig. 6b).

HLA-DR beta gene alleles are located on chromosome 6; *DRB1* is constitutively expressed in all individuals, and the presence of *DRB3, 4,* and *5* varies with each individual[14]. We performed *HLA-DRB1* and *DRB5* genotyping on an additional cohort of 209 treatment naïve RA patients to further characterize the contribution of the *HLA-DRB5* genotype to the development of different RA subtypes. Although the *HLA-DRB5* genotype was not significantly more prevalent in ACPA+ RA (49/141, 34.8%) compared with

ACPA- RA (15/48, 31.3%) and HCs (10/34,29.4%) (Supplementary Fig. 8a), evidence of disease activity, including DAS28, CRP, and IgM were significantly higher in ACPA+ RA with *HLA-DRB5* genotype, whilst this was not observed in ACPA- RA (Supplementary Fig. 8b). These results indicate a close association of *HLA-DRB5* genotype with disease activity in ACPA+ RA.

**Heterogeneous subpopulations of T and NK cell subsets in ACPA- RA.** CD8 T cells could be separated into four smaller clusters (Fig. 7a, b). The proportion of naïve CD8 T cells was decreased but cytotoxic CD8 T cells was increased in ACPA+ RA (Fig. 7c, Fig. 8a). CD4 T cells could be divided into nine subsets including NK, γδ T, and seven CD4 T cells subsets (Fig. 7a, b). RA patients displayed higher proportions of *S100A8^hi^GZMB^+* effector CD4 T (Teff) cells expressing cytotoxic effector CD4 T cell genes (including *GNLY* and *GZMB*) compared with HCs (Fig. 7c, Fig. 8a). The abundance of *GZMK^+* IFN-act central memory T (TCM) cells expressing central memory CD4 T cell and interferon pathway genes (including *GZMK, ISG15, IFI6, CCR7, GPR183, SELL, TCF7,* and *IL7R*) was also higher in RA patients, especially in ACPA- RA, compared to HCs (Fig. 7c,

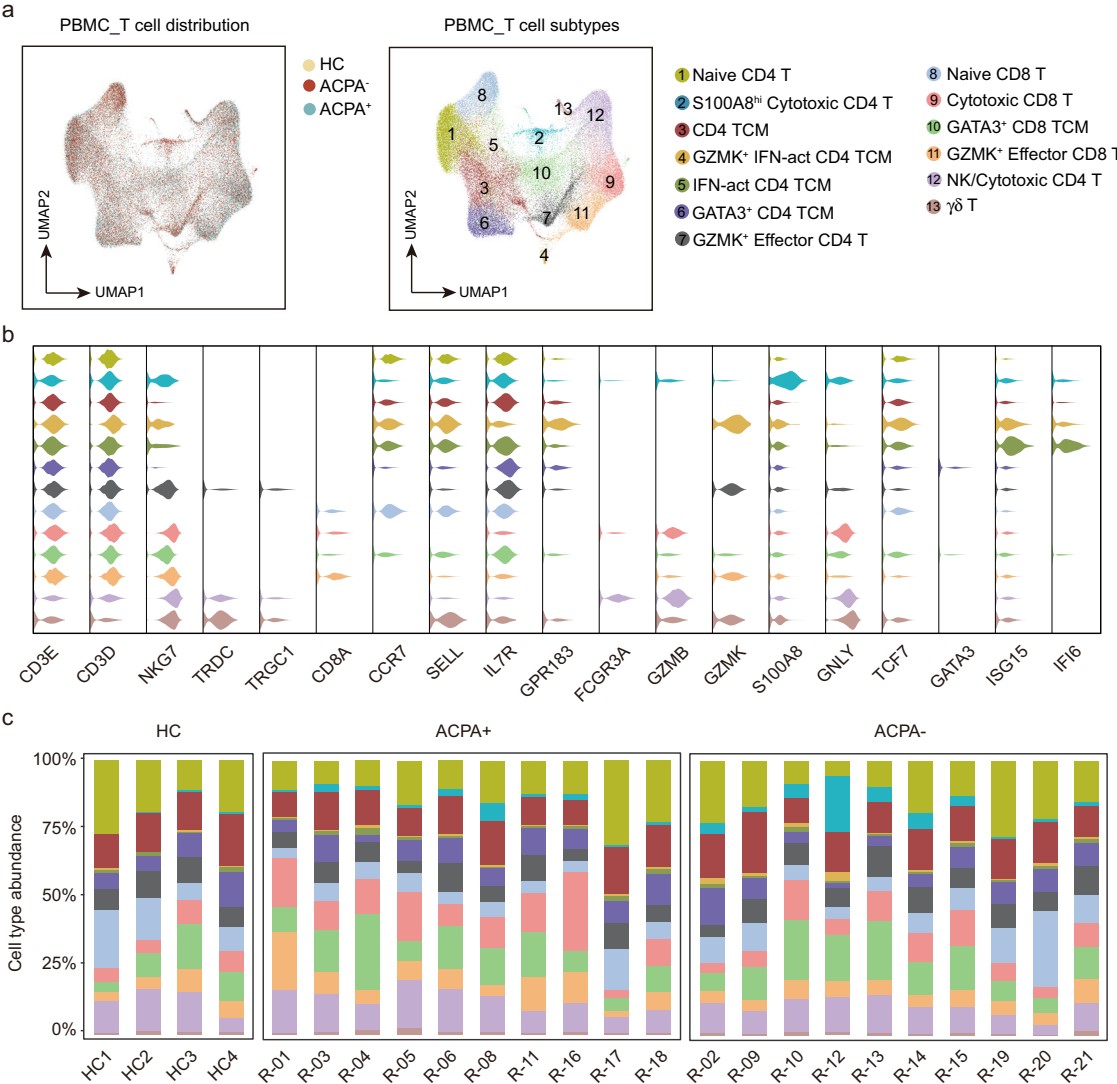

**Fig. 7 Heterogeneous subpopulations of T and NK cell subsets in the blood of patients with RA. a** UMAP visualization of T/NK cells from PBMC, with 13 cell subtypes identified across 70,650 cells. Cells are marked by ACPA type(left) and cell subtypes (right). **b** Violin plots showing marker genes across PBMC T/NK cell subtypes in a. The y axis represents log-scaled normalized counts. **c** Bar plots showing the relative percentage of T/NK cell subtypes in PBMC for each sample.

Fig. 8a). GO and KEGG pathway analysis showed that $S100A8^{hi}GZMB^{+}$ Teff cells expressed gene signatures that were enriched for "osteoclast differentiation" and "T cell receptor signaling pathway" compared with canonical cytotoxic NK/CD4 CTL subsets (Fig. 8b). Besides, the cytotoxic signature gene expressions, including *GZMB*, *NKG7*, and *PFN1* was significantly lower in cytotoxic cells in PBMC of ACPA- RA (Fig. 8c).

Synovial T cells could be sub-clustered into eight subsets based on typical gene expressions, including central memory T (TCM), effector memory T (TEM) (*GPR183*, *SELL*, *TCF7*, and *IL7R*), exhausted CD4 T (*PDCD1*, *LAG3*, and *TIGIT*), Treg (*FOXP3*), cytotoxic CD8 T (*CD8A*, *GZMB*, *NKG7*, and *CD3D*), NK (*GZMB*, *NKG7*), NKT/γδ T (*TRDC*, *TRGC1*, *NKG7*, and *CD3D*) and ILC3 (*RORC*, *IL23R*, and *IL7R*) (Fig. 9a-b). Notably, the proportions of NK cells and exhausted CD4 T cells were lower in ST of ACPA-RA (Fig. 9c). Besides, the cytotoxic signature gene expressions, including *GZMB*, *NKG7*, and *PFN1* was also significantly lower in cytotoxic cells in ST of ACPA- RA (Fig. 9d). Moreover, the exhausted signature gene expressions, including *LAG3* and *PDCD1* were absent in exhausted CD4 T and CD8 TEM of ACPA- RA ST T cells (Fig. 9d). Interestingly, a distinct high

expression of *MMP3* in T cells was found in ST of ACPA- RA (Fig. 9d). Together, these results indicate the decreased cytotoxic features in the peripheral blood and ST of ACPA- RA patients.

**Interaction between macrophage subsets, DCs, and CD4⁺ T cell subsets**. To further investigate the interactions between macrophage subsets, DCs, and CD4 T cells in RA, we utilized CellPhoneDB to better understand the regulatory relationships among these cell clusters. While the interactions of *HLA-DRB5⁺* DC and *HLA-DRB5⁻* DC with *S100A8^{hi}* Cytotoxic CD4 T or NK/CD4 T were similar, *HLA-DRB5⁻* pDC, which were more abundant in ACPA- RA, exhibited stronger interactions with cytotoxic T subsets than *HLA-DRB5⁺* pDC (Fig. 10a). Both *HLA-DRB5⁻* $CCL^{+}$ and *HLA-DRB5^{hi}CD36⁺* macrophages exhibited stronger interactions with T/NK subsets in ST than $CCL^{+}$ $IL18^{+}$ macrophages (Fig. 10b).

We further utilized a set of immune-related ligand-receptor (L-R) pairs to predict those involved in these cell-cell interactions. DC subtypes employed CD74-APP/COPA/MIF to interact with Teff cell subtypes (Fig. 10c). *HLA-DRB5⁻CCL⁺* macrophages,

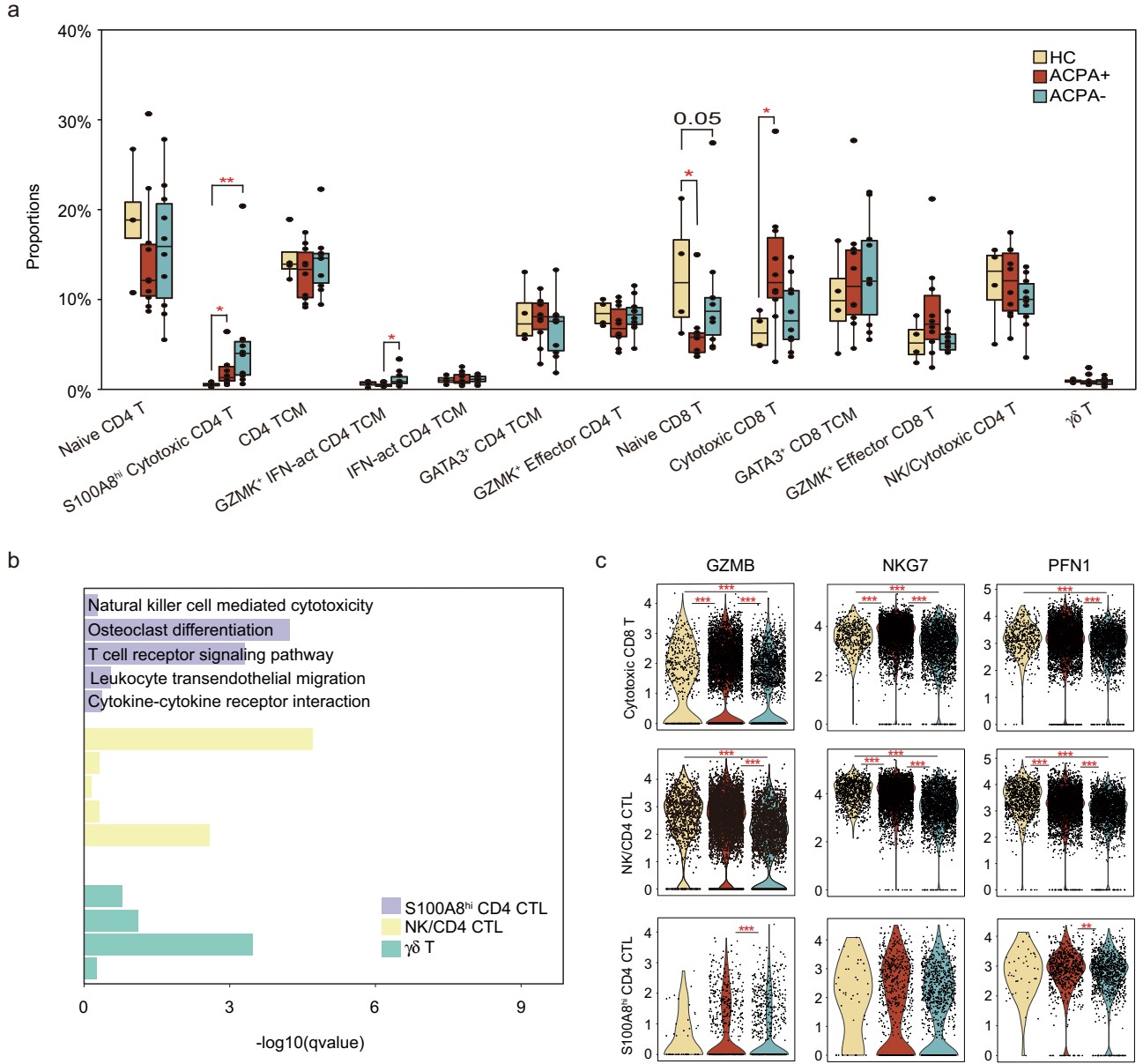

**Fig. 8 Expression pattern of T and NK cell subsets in the blood of patients with RA. a** Box plots showing the proportions of each T/NK cell subtype in PBMC across ACPA groups. Cell types showed enrichment in ACPA+ or ACPA- subgroups are marked with *. P values were calculated by the two-sided Wilcoxon test. *p < 0.05 (from left to right, p = 0,014, 0.011, 0.024, and 0.036), **p < 0.01 (p = 0.004). n = 4 for HC, n = 10 for ACPA+ group, and n = 10 for ACPA- group. **b** Bar plots showing the enriched upregulated KEGG pathways and GO biological processes in terms of specific T/NK cell subtypes. **c** Violin plots showing the differences in interested gene expression between ACPA- and ACPA+ T/NK cell subtypes in PBMC. Asterisks indicate the significance. P values were calculated by the two-sided Wilcoxon test. *p < 0.05, **p < 0.01, ***p < 0.001.

which were significantly upregulated in the ST of ACPA- RA, was more prone to utilize CCR8-CCL18, CCR1-CCL18, CCR1-CCL13, and CCR2-CCL13 to interact with ST T subsets (Fig. 10d). ST T subsets interact with macrophage subsets via CCL3-CCR1, CCL3-IDE, and CCL3-CCR5 (Fig. 10e). Altogether, our interaction data reveal that abnormal DC, T cell, and macrophage subsets present in RA patients display a stronger interaction with each other via the CD74, CCL13, CCL18, and CCL3 molecules, which could confer signals contributing to the abnormal immune/inflammatory responses in RA.

## Discussion
The immune response against citrullinated antigens is a hallmark of ACPA+ patients with RA, although it remains uncertain

whether RA can be truly divided into two clinical subtypes with distinct mechanisms of immune-dysregulation based solely on serum ACPA status. To our knowledge, this study is the first to comprehensively address differences in the distribution of CD45+ immune cell subsets by single-cell RNA sequencing in both the peripheral blood and synovial tissue, and systematically compare them between ACPA- and ACPA+ RA.

In ACPA- RA, we observed several specific abnormalities in the synovial membrane (Supplementary Table 3), including significant upregulation of *CCL13*, *CCL18* expression in B, DC, and macrophage subsets, and upregulation of *MMP3* expression in DC, macrophage, and T cells subsets, respectively. We also observed a lack of *HLA-DRB5* expression on B, DC and macrophage subsets, and decreased cytotoxic T cells and exhausted

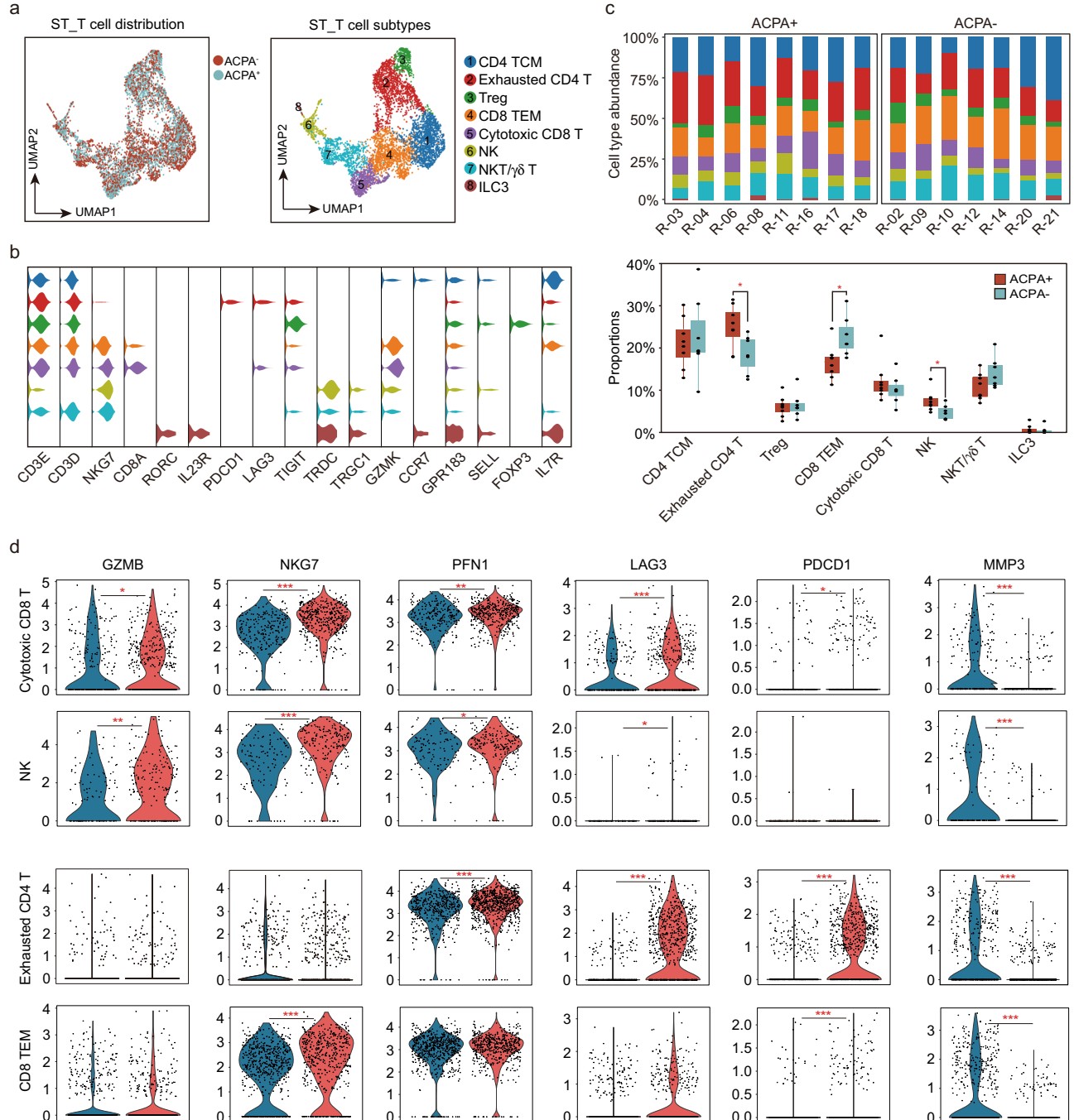

**Fig. 9 Identification of synovial T and NK cell subsets in patients with RA. a** UMAP visualization of T/NK cells from STMC, with eight-cell subtypes identified across 5760 cells. **b** Violin plots showing marker genes across STMC T/NK cell subtypes in F. The **y** axis represents log-scaled normalized counts. **c** Top: bar plots showing the relative percentage of T/NK cell subtypes in STMC for each sample. Bottom: box plots showing the proportions of each T/NK cell subtype in STMC across ACPA groups. Cell types showed enrichment in ACPA+ or ACPA- subgroups are marked with *. *P* values were calculated by the two-side Wilcoxon test. *p < 0.05 (from left to right, p = 0.029, 0.014, 0.021). n = 10 for ACPA+ group, and n = 10 for ACPA- group. **d** Violin plots showing the differences in interested gene expression between ACPA- and ACPA+ T/NK cell subtypes in STMC. Asterisks indicate the significance. *P* values were calculated by the two-sided Wilcoxon test. *p < 0.05, **p < 0.01, ***p < 0.001.

T cells in the synovial membrane of ACPA- RA. This constellation of abnormally distributed immune/inflammatory cells suggests that the local synovial response is substantially different in ACPA- and ACPA+ RA.

We observed significantly upregulated expression of *CCL13*, *CCL18*, and *MMP3*. CCL13 was found aberrantly expressed in the serum, synovial fluid, and synovial tissues from RA patients compared with those with osteoarthropathy, which could

enhance fibroblast-like synoviocytes (FLS) proliferation, increase macrophage infiltration, and synovial tissue angiogenesis[15,16]. CCL18 levels in serum and synovial tissues were found to be elevated in RA patients compared with controls, and were associated with RA disease progression[17,18]. MMP3 was found to be elevated in the serum of RA and associated with bone erosion development[19]. We found that CCL+ macrophages were more prone to utilize CCR8-CCL18, CCR1-CCL18, CCR1-CCL13, and

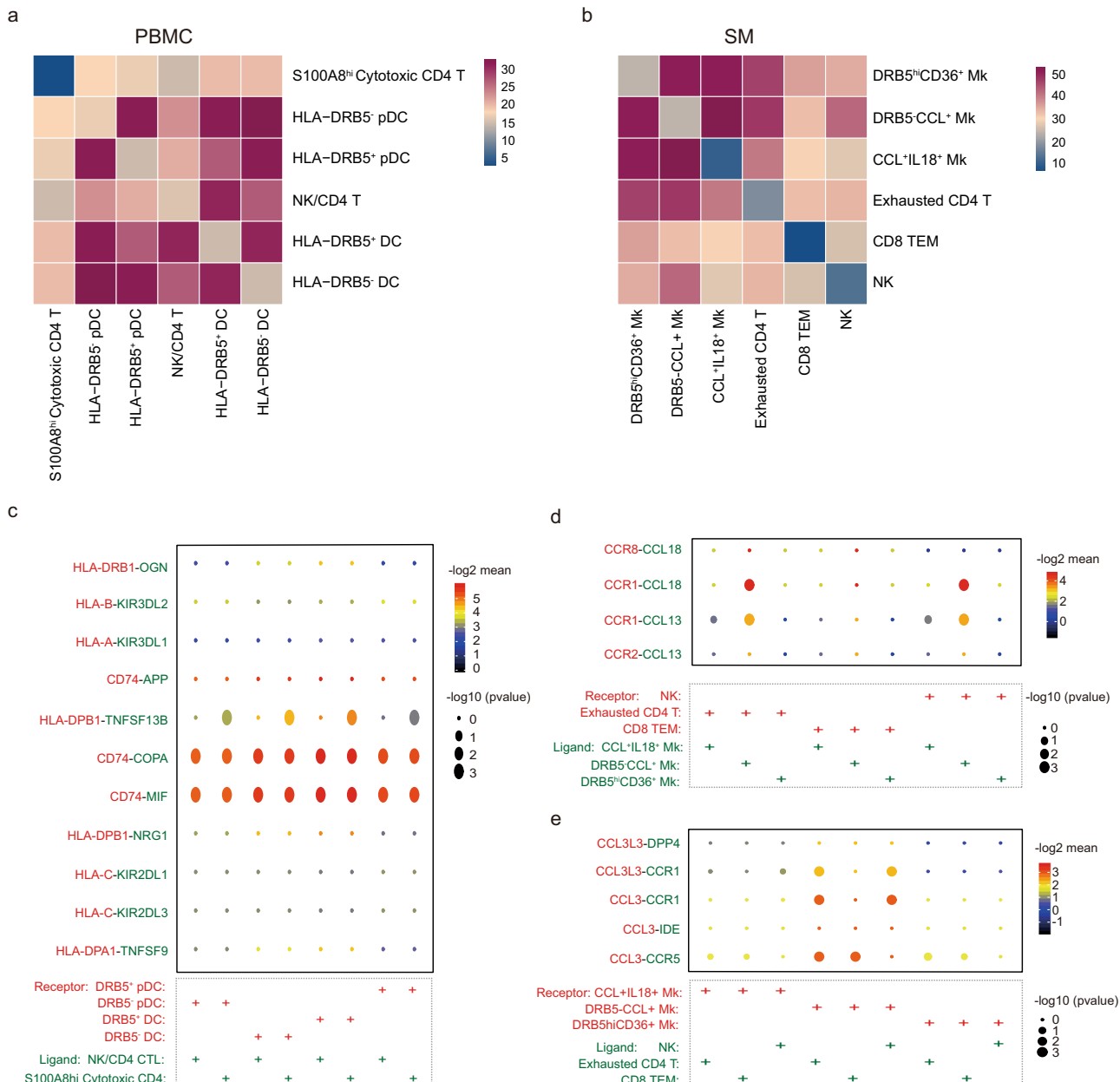

**Fig. 10 Interaction relationships between macrophage, DC, T cell, and NK cell subsets. a** Heatmap displaying the key cell-cell interaction pairs in PBMC. **b** Heatmap displaying the key cell-cell interaction pairs in the synovium. **c** Dot plot showing the significant ligand-receptor pairs involved in the interaction between DC subsets and CD4 T cell subsets. **d** Dot plot showing the significant ligand-receptor pairs involved in the interaction between macrophage subsets and T cell or NK subsets. (Ligand: T cell or NK subsets, Receptor: Macrophage subsets). **e** Dot plot showing the significant ligand-receptor pairs involved in the interaction between T cell or NK subsets and macrophage subsets. (Ligand: Macrophage subsets, Receptor: T cell or NK subsets).

CCR2-CL13 to interact with ST T subsets. Notably, CCL18 has been reported to inhibit CCR1 mediated chemotaxis[20], which might explain the observed significantly lower proportions of NK and exhausted CD4 T cells in ST of ACPA- RA.

Lewis et al. have reported clustering of patients into three categories as *lympho-myeloid* (CD20 B cell aggregate rich), *diffuse-myeloid* (CD68 rich in the lining or sublining layer but poor in B cells), and *pauci-immune* (with scant infiltration of immune cells and predominantly of FLS)[21]. Synovial RNA-seq comparing differential gene expression between ACPA+ and ACPA- RA patients showed increased plasma cell genes such as *XBP1*, *ODC1*, and *EAF2*, indicating a pro-lympho-myeloid pathotype in ACPA+ RA[21]. We further compared gene

expression in different ST-derived plasma B between ACPA- and ACPA+ RA. Consistent with Lewis' findings, we observed increased *ODC1* expression in *HLA-DRB5+* memory B and *IGLL5+* Plasma B (Supplementary Fig. 9). Moreover, we also found the absence of exhausted T cell markers and decreased cytotoxic T/NK cells in ST of ACPA- RA, further supporting that ACPA- RA patients are more likely clustered into *diffuse-myeloid* or *pauci-immune* category. Humby et al. also revealed that *diffuse-myeloid* or *pauci-immune* RA were less likely to develop joint damage progression than *lympho-myeloid pathotype*[22]. However, the *pauci-immune* RA patients predicted inadequate response to TNF blockade treatment[23]. Our present study demonstrates that although exhibiting similar clinical arthritic

symptom, the immune pathogenesis of ACPA- and ACPA+ RA patients is quite different, which contributes to the histological, disease progression, and therapeutic strategy heterogeneity.

Our study demonstrated a close association of the *HLA-DRB5* genotype with disease activity in ACPA+ RA. Human leukocyte antigen-DR15 is a haplotype that results in the expression of *DRB1* and *DRB5*, most frequently HLA-DRB1*15:01 and HLA-DRB5*01:01. Previous studies demonstrated the association of the HLA-DR15 haplotype with multiple sclerosis, a chronic T cell-mediated autoimmune disease[24,25]. HLA-DR15 status was also reported as an important marker in autoimmune bone marrow disease[26]. The study of the role of HLA-DR15 in multiple sclerosis demonstrated that patients displayed T cell proliferative responses to myelin-associated autoantigens, which was related to the presence and level of HLA-DR15 expression[27]. In addition, a recent study described autoproliferation of peripheral Th1 cells mediated by memory B cells in an HLA-DR15 haplotype-dependent manner[28,29]. In this study, we observed the lack of *HLA-DRB5* expression and lower antigen-presenting ability in ST memory B cell subsets, as well as macrophage and DC subsets of ACPA- RA. Therefore, we speculate that autoimmune B and T cell subsets are not major players in the pathogenesis of synovium phenotype of ACPA- RA. The presence of *HLA-DRB5* genotype in ACPA+ RA might also indicate autoproliferation of Th1 cells to present autoantigen, contributing to disease activity. Furthermore, the absence of an "intermediate" *HLA-DRB5*+ Plasma B, higher expression of *IGHM*, and lower expressions of *IGHG3* in the peripheral blood of ACPA- RA, indicated the abnormal class-switching of B cell subsets in ACPA- RA.

All samples included were taken at the time of arthroplasty, indicating the potential contribution of OA to clinical symptom in our present study. Although synovitis is a common feature of OA and characterized by immune cells infiltration, a recent review summarized the published studies regarding the presence of inflammatory cells and their cytokines in OA ST[30], and suggested that the number of infiltrating cells as well as cytokine were significantly lower in OA than in RA ST[30]. Moreover, a previous study performed by Raychaudhuri et al. applied scRNA-seq, mass cytometry, bulk RNA-seq, and flow cytometry to define inflammatory cell states in RA joint synovial tissues[12]. They reported that OA samples were characterized by a high abundance of fibroblast and endothelial cells but not significant in B cells, T cells, DC, or macrophages, as observed in ST between ACPA- and ACPA+ RA patients in our study.

Besides *HLA-DRB5*, CCL and *MMP3*, we observed other outstanding markers of macrophage subsets to be different between ACPA- and ACPA+ RA. We found that ACPA- RA ST macrophage subsets displayed upregulated *IL1B* expression (M1 macrophages) but decreased *TGFB1* (M2 macrophages) and *CD36* (phagocytic) expression. A previous study identified four transcriptionally distinct monocyte subsets in the scRNA-seq data: *IL1B*+ monocytes, *NURP1*+ monocytes, *C1QA*+ monocytes, and IFN-act *SPP1*+ monocytes, and leukocyte-rich RA have a greater abundance of *IL1B*+ monocytes and IFN-act *SPP1*+ monocytes[12].

Kurowska-Stolarska et al. used CD163, MerTK, and CD206 to define ST macrophages (STM) and found MerTK[pos]CD206[pos] STM were associated with remission maintenance[31]. They also performed scRNA-seq of MerTK[pos]CD206[pos] and MerTK[neg]CD206[neg] STMs, and identified nine distinct STM clusters. Of which, HLA[high]CLEC10A[high], TREM2[high]TIMD4[pos] and S100A12[pos] macrophages were consistent with *CLEC10A*+ macrophages, *TIMD4*+ *CCL*+ macrophages, *IL1B*[hi]*S100A12*[hi] macrophage identified in our present study, respectively. Intriguingly, they found that, in contrast to proinflammatory MerTK[neg]CD206[neg] STMs, MerTK[pos]CD206[pos] could induce FLS TGF-β response genes and a MerTK inhibitor increases FLS

expression of *MMPs1/3/14* and *IL-6*. With this study, we propose that ACPA- RA ST macrophage subsets might be MerTK[neg]CD206[neg] STM and ACPA+ RA ST macrophage subsets might be MerTK[pos]CD206[pos]. Should our proposition be validated by future studies, MerTK[neg]CD206[neg] STM might become a promising therapeutic target in the clinic.

In summary, our study provides a transcriptional landscape of immune cell status at single-cell resolution in ACPA- and ACPA+ RA. We identified the HLA-DR15 haplotype as a risk factor for developing the active disease in ACPA+ RA. We further demonstrated that significantly higher expression of *CCL13*, *CCL18*, and *MMP3* in DC and macrophage subsets, together with poor B cells and T cells responses are salient immune features in ST of ACPA- RA. Our findings suggest that ACPA- RA may preferentially utilize different immune mechanisms and pathways than ACPA+ RA. In ACPA- RA ST, inflammatory myeloid cells, including M1 macrophages (*IL1B*) and *MMP3* secreting DC subsets contribute to synovial pathogenesis while in ACPA+ RA ST, lymphoid cells (B cells and T cells) are major contributors. Therefore, they might respond to different therapeutic strategies.

## Methods

**Patient recruitment and ethics.** Peripheral blood and synovial membrane samples were obtained from patients who fulfilled the 2010 ACR/EULAR Rheumatoid Arthritis classification criteria and who underwent arthroplasty at Peking Union Medical College Hospital. Patient basic information as well as C-reactive protein, rheumatoid factor (RF), anti-CCP antibody status, erythrocyte sedimentation rate (ESR), disease duration, and DAS28 score were recorded and presented in Supplementary Table 1. Detailed information regarding patient HLA genotyping is presented in Supplementary Table 2. Informed consents were obtained from all human participants, and our study was approved by the Peking Union Medical College Hospital Ethics Committee (no. JS-1940).

**Sample preparation for 10× genomics.** Synovial membrane tissues were washed with cold PBS, cut into small pieces, and digested with 3 mg/ml collagenase IV, 1.5 mg/ml dispase II, and 0.1 mg/ml DNase I in PBS containing 10% FBS at 37 °C for 5 min. After three rounds of digestion, the cell supernatants were collected, combined, and passed through a 70-μm cell strainer. Mononuclear cells were isolated from peripheral blood and synovial membrane tissues using Ficoll-Paque gradient centrifugation (CL5020, CEDARLANE). CD45+ cells were isolated using magnetic beads (130-045-801, Miltenyi Biotech). Cell quantity and viability were assessed using Trypan Blue staining.

**10× genomics single-cell RNA sequencing.** Single-cell 3' gene expression libraries for CD45+ immune cells were prepared by strictly following the protocols of the Chromium Single Cell 3'v2 Library kit (10× Genomics). All resulting libraries were sequenced on the Illumina Xten PE-150 platform (ANOROAD and Novogene, Beijing, China).

**Single-cell RNA-seq data alignment and quality control.** Raw 10× Genomics sequencing data were processed using the CellRanger software v. 2.2.0, and the 10× human transcriptome GRCh38-1.2.0 was used as the reference. Single-cell read counts from all samples were first converted to individual Seurat objects using the Seurat (v3.1.1) analysis package in R (v3.6.1). For each object, we filtered data based on the number of unique molecular identifiers (UMI) and the number of genes detected. We retained genes expressed in at least five cells and cells in which the number of genes detected ranged from 500 to 3500. Each filtered Seurat object was then normalized using Seurat's NormalizaData function. Highly variable genes were detected using the FindVariableFeatures function in Seurat.

**Integration of scRNA-seq data from the same tissue.** All individual Seurat objects from the same tissue (PBMC or synovial tissue) were integrated using the FindIntegratedAnchors and IntegrateData functions in Seurat. Canonical correlations with highly variably expressed genes were then used for downstream dimension reduction and clustering analysis.

**Dimension reduction and major cell type annotation.** Peripheral blood mononuclear cells (PBMC) and synovial tissue objects were analyzed separately. The number of unique molecular identifiers, percentage of mitochondrial genes, and cell cycle genes were regressed out, and genes were scaled to unit variance. Principle component analysis (PCA) was performed. Clusters were then identified using Uniform Manifold Approximation and Projection (UMAP). Cell identity was assigned using known markers shown in Fig. 1C and Supplementary Figure 1A, C.

**Sub-clustering of B cells, DCs, monocytes, macrophages, and T Cells**. B cells, DC, monocytes, and macrophages, and T Cells were extracted from PBMC and synovial tissues for further sub-clustering. After extraction, genes were scaled to unit variance. PCA and clustering were performed as described in Dimension Reduction and Major Cell Type Annotation section. Doublet clusters were removed following the criteria described below: (1) B cell, DC, monocytes and macrophage subclusters in PBMC with CD3 expression (calculated as the mean expression of *CD3D*, *CD3E*, and *CD3G*) > 0.1; (2) B cell, DC, T cell subclusters in synovial with *CD14* expression > 0.1.

**Detection of differentially expressed genes and pathway analysis**. Differential gene expression (DEG) testing was performed using the FindMarkers function in Seurat with Wilcoxon test and p values were adjusted using Bonferroni correction. DEGs were filtered using a minimum $\log_2$(fold change) of 0.5 and a maximum adjusted p value of 0.05 and were then ranked by average $\log_2$(fold change) and FDR. Enrichment analysis for the functions of the DEGs was conducted using the clusterProfiler (v3.12.0) package[32] and DAVID 6.8[33,34]. The gene sets were based on Gene Ontology terms and Kyoto Encyclopedia of Genes and Genomes (KEGG) pathways.

**Trajectory analysis**. The Monocle 3 R package was used to construct single-cell developmental trajectories for B cells[35–38]. The raw expression matrices were fetched from the associated Seurat object using the GetAssayData function[39,40]. After creating the CellDataSet object in Monocle 3, data were normalized and preprocessed using the preprocess_cds function. Next, the dimensionality of the data was reduced using UMAP with the reduce_dimension function. Trajectory inference analysis was performed using the cluster_cells and learn_graph functions.

**Cell-cell interaction analysis**. To comprehensively analyze cell-cell interactions between immune cells, we used CellPhoneDB (v2.1.2)[41,42]. We derived potential ligand-receptor interactions based on the expression of a receptor by one cell subpopulation and ligand expression by another. We separately fetched the normalized counts of HCs, ACPA-positive, and ACPA-negative cells and used them as input for the CellPhoneDB algorithm.

**Data visualization**. All plots were generated using the ggplot2 (v 3.2.1), pheatmap (v 1.0.12), and EnhancedVolcano (v 1.2.0) packages in R 3.6.1. Box plots are defined as follows: the middle line corresponds to the median; the lower and upper hinges correspond to the first and third quartiles, respectively; the upper whisker extends from the hinge to the largest value no further than 1.5× the interquartile range (or the distance between the first and third quartiles) from the hinge; and the lower whisker extends from the hinge to the smallest value at most 1.5× the interquartile range from the hinge. Data beyond the end of the whiskers were designated as "outliers" and are plotted individually.

**Immunohistochemistry**. Synovial tissues were fixed in 4% paraformaldehyde overnight, dehydrated, and embedded in paraffin. 4-µm-thick sections were prepared, and antigen retrieval was conducted in citrate buffer (pH 6.0) at 95℃ for 10 min. Further staining and image scanning were conducted by ZSGB-Bio (Beijing, China). The following primary antibodies were used: anti-human MCP4 (CCL13) (ab224593, Abcam), anti-human CCL18 (ab233099, Abcam), and anti-human CCL3 (ab32609, Abcam).

**HLA genotyping**. *DRB1-DRB5* haplotypes were determined by BGI (Beijing Genomics Institute) using the super high-solution single-molecule sequence-based typing (SS-SBT) method[43]. In brief, genomic DNA was extracted from whole blood samples and sequencing was performed using an ABI3130 (Life Technologies, Carlsbad, CA).

**Reporting Summary**. Further information on research design is available in the Nature Research Reporting Summary linked to this article.

## Data availability

Sequence data have been deposited in the Genome Sequence Archive in BIG Data Center, Beijing Institute of Genomics (BIG), Chinese Academy of Sciences, under the accession code HRA000155. Source data are provided with this paper.

## Code availability

Custom scripts used in single-cell RNA-seq data analysis are available at: https://github.com/yeee904/RA_scRNA.

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

## Acknowledgements

This work was supported by grants from the National Natural Science Foundation of China (81788101, 81630044, 81971544, and 81801635), the Chinese Academy of Medical Science Innovation Fund for Medical Sciences (CIFMS 2017-12M-1-008, 2016-12M-1-003, 2017-I2M-3-011, and 2016-12M-1-008), Beijing Capital Health Development Fund(2020-2-4019), the Medical Epigenetics Research Center Fund, Chinese Academy of Medical Sciences (2017PT31035), National Science and Technology Major Project (2018ZX10302205, 2019YFC1315702), and the Guangdong Province Key Research and Development Program (2019B020226002).

## Author contributions

X.Z., F.B., and G.C.T conceived and supervised the project. X.W., M.W., and S.J. performed the experiments. Y.L., S.J., and X.W. performed bioinformatic analysis. X.Z., G.C.T, and P.E.L. wrote the manuscript with inputs from X.W., Y.L., S.J., H.L., and F.B.. B.Y., X.L, Y.J., X.J, M.W., and Y.Z. provided patient care and clinical and histological assessments. Y.F., H.Y., L.Z., and H.C. recruited patients for HLA genotyping. Y.Z. and X.W. processed clinical data. All authors discussed and approved the manuscript.

## Competing interests

The authors declare no competing interests.
