## [Peer Review File · Nature Communications]

REVIEWER COMMENTS

Reviewer #1 (Remarks to the Author):

General overview: The authors convincingly document that the cellular composition of synovial tissues from ANCA+ RA patients differ from the cellular composition of synovial tissues from ANCA- RA patients. (This is not especially surprising, given the considerable phenotypic differences between ANCA+ and ANCA- RA patients.) The study is strictly descriptive, with no experiments geared to elucidating mechanistic underpinnings.

Specific concern: None of the 14 RA patients were taking DMARDs, steroids or targeted therapies at the time of sampling -- why not? If they were clinically active, why were they not being treated? If they were not clinically active, then the results obtained may not necessarily be germane to "active" RA.

Reviewer #2 (Remarks to the Author):

This manuscript employs scRNA-seq to examine PB controls and PB and ST of patients with ACPA+ and ACPA- RA. The data identified a variety of subsets of B and T cells, DCs and monocytes/macrophages. The proportions of the subsets were compared between the groups and programs were used for trajectory analysis and to identify potential cell-cell interactions. The data suggest that HLA-DR5+ memory B cells with potential interactions with GZMB effector T cells were a feature of ACPA+ ST cells. HLA-DR5+ macrophages were also enriched in ACPA+ ST, whereas CCL+ memory B cells, pDC and macrophages were enriched in ACPA- ST.

Major Concerns:

1. The results states that the patients were off medications at the time the samples were obtained. This is hard to understand since the samples were obtained at the time of arthroplasty. Further, sup table 1 identifies medications the patients were on. If the medications were stopped prior to the surgery for how long and was it medically justified.
2. Since the samples were taken at the time of arthroplasty, what is the contribution of OA to the changes observed. This should at least be discussed.
3. The presentation of the data is very repetitive, making it difficult to focus on critical information. It is extremely descriptive.
4. In panel D of Figures 2-4 in the PB and SM, ACPA- is represented by different colors, adding to the difficulty of reading the manuscript.
5. Many of the conclusions are "tendency", "almost absent", "more abundant" so it is hard to keep track of what is statistically documented and what is wished for.
6. An example lines 180-1: "Peripheral plasma cells from ACPA- RA did not display a tendency to synovial migration as that observed in ACPA+ RA". Why do the authors assume the plasma cells migrated, they may have differentiated within the joints.
7. Line 232: indicated CCL+ pDC were enriched in ACPA- RA, but to my reading it was ACPA+ RA.
8. Line 317: DR5+ memory B cell were "almost absent" in SM of ACPA+ RA, where as Fig 2 shows they were increased.
9. Line 367: DR5+ MK mainly resided in SM of ACPA+ RA, while Fig 4d shows the opposite.

Reviewer #3 (Remarks to the Author):

This is an impressive tour de force with a very high number of single cells from three populations of individuals being subject to single cell RNA sequencing. Despite this extraordinary amount of sequencing the number of individuals (and specimens) from each group remains rather small, and this limited number of specimens contrasts with the very large numbers of analyses and comparisons that are presented. Further, comparisons of synovial tissue-derived cell phenotypes

between ACPA+ and ACPA- patients is further complicated from knowledge recently provided in some detail, that the structure and cell composition of synovial membranes may differ a lot between synovial tissues within the groups of ACPA+ and ACPA-negative patients. Taken together these problems and limitations of the study should be better recognized and commented on by the authors, and conclusions concerning the differences between ACPA+ and ACPA- patients be more cautious. Nevertheless, the extensive data on RNAseq on carefully characterized subpopulations of blood and synovial cells from RA patients represent an important piece of data of value for the scientific community, and the presentations of these data would benefit from a better recognition of the limitations from low numbers of specimens and high numbers of comparisons.

In addition, I have some more specific comments:

- A better description of the characteristics of the synovial tissues that were analyzed would be beneficial in order to try to fit them into the characteristics of synovial biopsies that have recently been described by Pitzalis and others.
- A major section addresses differences between ACPA+ and ACPA- patients concerning B cells from blood and synovial tissues. Here, the authors choose to “integrate” the analysis of B cells from peripheral blood and synovial tissue. This integration harbors some problems as the phenotypes of B cells from blood and synovial tissue are quite different, and there is no complete comparisons between blood on one hand and synovial tissue on the other (materials are small). A better rationale for this “integration” and discussion of differences between blood and synovial tissues is warranted.
- The comparison of the clinical course in ACPA-positive and ACPA negative patients regarding the influence of DRB5 is interesting, but numbers when dividing into four groups and comparing clinical course with the clinical disease activity outcome, is clearly underpowered, and obviously not subjected to any verification in other data sets. This piece is, in the eyes of this reviewer, the weakest part of the paper and if evaluated in its own right, would require much larger materials and verification in independent datasets. In this reviewer’s opinion, this part may be deleted from the paper without reducing the impact of the valuable information from the massive single cell sequencing.

A point-to-point response to reviewers' comments

Reviewer #1 (Remarks to the Author):

General overview: The authors convincingly document that the cellular composition of synovial tissues from ANCA+ RA patients differ from the cellular composition of synovial tissues from ANCA- RA patients. (This is not especially surprising, given the considerable phenotypic differences between ANCA+ and ANCA- RA patients.) The study is strictly descriptive, with no experiments geared to elucidating mechanistic underpinnings.

Response: The presence of ACPA in serum has enabled the diagnosis, treatment decision making and follow-up of RA patients. Currently, great efforts are underway to help understand the contribution of ACPA to bone loss and the development of joint destruction, whereas ACPA- RA has been relatively less investigated. The uncertainty in the diagnosis of ACPA- RA and a conservative approach to the treatment selection of ACPA- RA due to the concerns about potential drug toxicity often lead to an inappropriate or delayed therapy.

ACPA- RA cannot simply be considered as a mild form of RA, instead it could represent a completely different subgroup of RA. However, our knowledge of the immunological mechanisms involved in the expression of ACPA- RA is still very limited. Although our study may appear descriptive, we would like to emphasize that this is the first to apply scRNA-seq to specifically and systematically characterize the immune cell composition, proportion, gene expression signature, and development trajectory of CD45⁺ cells in the peripheral blood and synovial tissues of ACPA- RA patients. We consider this is the major contribution of our work to the field. The extensive data on scRNA-seq carefully characterized subpopulations of peripheral blood and synovial cells from RA patients represent an important piece of data of value for the scientific community.

To confirm the scientific value and contributions, in the revised version, we have made tremendous effort to enroll additional patients for scRNA-seq and *HLA-DRB5* genotyping, and we also separated and assorted the scRNA-seq data by peripheral blood and synovial tissues, reanalyzed all the cell subsets to make the analysis more accurate and easier to focus on. Consistently, we observed several unique and important abnormalities in synovial tissues, including a significantly up-regulation of CCL13, CCL18 in B, DC and Mk cell subsets, and MMP3 expression in DC, Mk and T cell subsets, respectively.

In addition, we have performed immunohistochemical staining experiments to validate the observed increased infiltration of CCL13 and CCL18-expressing immune cells in synovial tissues of ACPA- RA patients. We also observed lack of *HLA-DRB5* on B, DC and Mk cell subsets, and decreased cytotoxic T/NK cells and exhausted T cells in the synovial membrane of ACPA- RA patients. We further performed *HLA-DRB5* genotyping of 209 treatment-naïve RA patients and demonstrated that the

HLA-DR15 haplotype (*DRB1/DRB5*) conveys an increased risk of developing active disease in ACPA+ RA.

Specific concern: None of the 14 RA patients were taking DMARDs, steroids or targeted therapies at the time of sampling -- why not? If they were clinically active, why were they not being treated? If they were not clinically active, then the results obtained may not necessarily be germane to "active" RA.

Response: We thank the reviewer for his insightful question and this is exactly one of the points that highlights the importance and significance of our study. Although unusual now in most rheumatology practices in China, there are still some patients who have established disease who are not taking DMARDs, steroids or targeted therapies. This may be due to a variety of reasons including limited access to a rheumatologist, poor education, etc. Although some patients were clinically active, they choose to use physical therapies (such as heat incubation or acupuncture) to alleviate their pain without taking DMARDs, steroids or targeted therapies. In Peking Union Medical College Hospital (PUMCH), we treat more than 30,000 RA patients per year, and in this study, we managed to screen RA patients that were not-treated or off medical treatment for at least three months when their samples were collected to exclude the interference of past DMARDs and to unveil the bona fide nature of the ACPA- RA.

Reviewer #2 (Remarks to the Author):

This manuscript employs scRNA-seq to examine PB controls and PB and ST of patients with ACPA+ and ACPA- RA. The data identified a variety of subsets of B and T cells, DCs and monocytes/macrophages. The proportions of the subsets were compared between the groups and programs were used for trajectory analysis and to identify potential cell-cell interactions. The data suggest that HLA_DR5+ memory B cells with potential interactions with GZMB effector T cells were a feature of ACPA+ ST cells. HLA-DR5+ macrophages were also enriched in ACPA+ ST, whereas CCL+ memory B cells, pDC and macrophages were enriched in ACPA- ST.

Response: Thank you very much for providing a succinct summary of our work.

Major Concerns:

1. The results states that the patients were off medications at the time the samples were obtained. This is hard to understand since the samples were obtained at the time of arthroplasty. Further, sup table 1 identifies medications the patients were on. If the medications were stopped prior to the surgery for how long and was it medically justified.

Response: The reviewer raises most obvious points. The fact that we were able to study patients with advanced RA and still treatment naïve represents

one of the points that highlights the importance and significance of our study. Although unusual now in most rheumatology practices in China, there are still some patients who have established disease yet are not taking DMARDs, steroids or targeted therapies. These may be due to a variety of reasons including limited access to a rheumatologist, poor education, etc. Although some patients were clinically active, they choose to use physical therapies (such as heat incubation or acupuncture) to alleviate their pain without taking DMARDs, steroids or targeted therapies. In Peking Union Medical College Hospital (PUMCH), we treat more than 30,000 RA patients per year, and in this study, we managed to screen RA patients that were not-treated or off medical treatment for at least three months when their samples were collected to exclude the interference of past DMARDs and to unveil the bona fide nature of the ACPA- RA.

The medications listed in sup Table 1 of the original manuscript showed the treatment received by each patient AFTER arthroplasty. In our revised manuscript, we have deleted the medication record to avoid confusion.

2. Since the samples were taken at the time of arthroplasty, what is the contribution of OA to the changes observed. This should at least be discussed.

Response: Thank you for the helpful suggestion. Indeed, the developing OA may alter the developing inflammatory process yet it is difficult to accurately address this point. We have discussed this point accordingly in our revised manuscript (paragraph 6, page 11).

3. The presentation of the data is very repetitive, making it difficult to focus on critical information. It is extremely descriptive.

Response: To improve the clarity and make the analysis more accurate and easier to focus on, we separated and assorted all the subsets analysis by peripheral blood and synovial tissue in our revised version. In brief, we observed a number of unique abnormalities in the synovial membrane, including significantly up-regulation of *CCL13*, *CCL18* in B, DC and Mk cells subsets, and *MMP3* expression in DC, Mk and T cells subsets, respectively. We also observed lack of *HLA-DRB5* on B, DC and Mk subsets, and decreased cytotoxic T/NK cells and exhausted T cells in the synovial membrane of ACPA- RA. We also summarized the critical information in Table S3 in the revised manuscript.

4. In panel D of Figures 2-4 in the PB and SM, ACPA- is represented by different colors, adding to the difficulty of reading the manuscript.

Response: Thank you for the helpful comment. We have used the same color of ACPA- in the PBMC and SM accordingly throughout the revised manuscript.

5. Many of the conclusions are “tendency”, “almost absent”, “more abundant”

so it is hard to keep track of what is statistically documented and what is wished for.

Response: According to the reviewer's suggestion, we assorted the analysis by peripheral blood and synovial tissue and clarified all those conclusions that were statistically significant throughout the revised manuscript.

6. An example lines 180-1: "Peripheral plasma cells from ACPA- RA did not display a tendency to synovial migration as that observed in ACPA+ RA". Why do the authors assume the plasma cells migrated, they may have differentiated within the joints.

Response: We agreed with the stated comment that the plasma cells could also might have differentiated within the joints from ST-resident memory B cells. This sentence has been deleted in our revised manuscript.

7. Line 232: indicated CCL+ pDC were enriched in ACPA- RA, but to my reading it was ACPA+ RA.

8. Line 317: DR5+ memory B cell were "almost absent" in SM of ACPA+ RA, whereas Fig 2 shows they were increased.

9. Line 367: DR5+ MK mainly resided in SM of ACPA+ RA, while Fig 4d shows the opposite.

Response: We apologize for these typos and mistakes, and we have carefully examined and corrected similar mistakes in the revised manuscript. Thank you very much for the careful reading and review.

Reviewer #3 (Remarks to the Author):

*This is an impressive **tour de force** with a very high number of single cells from three populations of individuals being subject to single cell RNA sequencing. Despite this extraordinary amount of sequencing the number of individuals (and specimens) from each group remains rather small, and this limited number of specimens contrasts with the very large numbers of analyses and comparisons that are presented. Further, comparisons of synovial tissue-derived cell phenotypes between ACPA+ and ACPA- patients is further complicated from knowledge recently provided in some detail, that the structure and cell composition of synovial membranes may differ a lot between synovial tissues within the groups of ACPA+ and ACPA-negative patients. Taken together these problems and limitations of the study should be better recognized and commented on by the authors, and conclusions concerning the differences between APAS+ and ACPA- patients be more cautious. Nevertheless, the extensive data on RNAseq on carefully characterized subpopulations of blood and synovial cells from RA patients represent an important piece of data of value for the scientific community, and the presentations of these data would benefit from a better recognition of the limitations from low numbers of specimens and high numbers of comparisons.*

Response: Thank you very much your high appraisal of our work. According to the reviewer's kind suggestion, to further verify the conclusions, in our revised manuscript we have included three more ACPA+RA and three more ACPA- RA patients for scRNA-seq and additional 82 treatment-naïve RA patients for *HLA-DRB5* genotyping.

In addition, I have some more specific comments:

- *A better description of the characteristics of the synovial tissues that were analyzed would be beneficial in order to try to fit them into the characteristics of synovial biopsies that have recently been described by Pitzalis and others.*

Response: Thank you for your constructive suggestion. This is a very important suggestion. We have performed the comparison of gene expression in different ST plasma B between ACPA- and APCA+ RA (please see **Figure below**). We found increased *ODC1* expression in *HLA-DRB5*⁺ memory B and *IGLL5*⁺ plasma B in ACPA+RA patients. Also, our study found absence of exhausted T cells markers and decreased cytotoxic T/NK cells in ST of ACPA-RA, further supporting that ACPA- RA patients are more likely clustered into *diffused-myeloid* or *pauci-immune* category that have recently been described by Pitzalis *et.al*. We have added these observations and analysis to the Discussion section (paragraph 4, page 36-37) of our revised manuscript, and related references were included (references #21-23). We appreciate your constructive suggestions which have really helped to improve the depth of our analysis and provide the evidence supporting different treatment strategies in clinic.

A.

B.

Figure. Differentially expressed genes in synovial B cell subtypes between ACPA- and ACPA+ RA patients

A. Volcano plots showing differentially expressed genes (DEGs) of *HLA-DRB5*⁺ Memory B cells in ACPA+ vs. ACPA- RA patients.

B. Volcano plots showing differentially expressed genes (DEGs) of *IGLL5*⁺ Plasma B cells in ACPA+ vs. ACPA- RA patients.

- A major section addresses differences between ACPA+ and ACPA- patients concerning B cells from blood and synovial tissues. Here, the authors choose to “integrate” the analysis of B cells from peripheral blood and synovial tissue. This integration harbors some problems as the phenotypes of B cells from blood and synovial tissue are quite different, and there is no complete comparisons between blood on one hand and synovial tissue on the other (materials are small). A better rationale for this “integration” and discussion of differences between blood and synovial tissues is warranted.

Response: Thank you very much for the kind and helpful suggestions. This is a very important suggestion. In our revised manuscript, following the reviewer’s suggestion, in addition to increase the samples, we have also separated and assorted the scRNA-seq data by peripheral blood and synovial tissues, and re-analyzed all the cell subsets separately to make the analysis more accurate and to improve the clarity of differential gene expression between ACPA- and ACPA+ RA patients.

- The comparison of the clinical course in ACPA-positive and ACPA negative patients regarding the influence of *DRB5* is interesting, but numbers when dividing into four groups and comparing clinical course with the clinical disease activity outcome, is clearly underpowered, and obviously not subjected to any verification in other data sets. This piece is, in the eyes of this reviewer, the weakest part of the paper and if evaluated in its own right, would require much larger materials and verification in independent datasets. In this reviewer’s opinion, this part may be deleted from the paper without reducing the impact of the valuable information from the massive single cell sequencing.

Response: Thank you very much for the kind suggestion. In our revised manuscript, we managed to collect additional 82 treatment-naïve RA patients

for *HLA-DRB5* genotyping. The conclusion that a close association of *HLA-DRB5* genotype with disease activity in ACPA+ RA is reinforced, implicating that the presence of *HLA-DRB5* genotype in ACPA+ RA might indicate auto-proliferation of Th1 cells to present autoantigen and contribute to disease activity.

REVIEWERS' COMMENTS

Reviewer #2 (Remarks to the Author):

The amount of data and the number of associations is very substantial. Supplemental table 3 is helpful at summarizing the data and providing potential mechanistic insights.

I reread the discussion hoping for insights into mechanistic differences between ACPA+/- RA, but was not rewarded. Much repetition of the data. When discussing the different histologic subsets described by others, inferences were made. However characterization of the histologic data of the samples used in this study was not made. Figure 4e is just an example.

I believe that a sentence or two should be included in the introduction to better explain the circumstances surrounding the acquisition of the tissue should be provided. It would be helpful for the reader to understand.

Reviewer #3 (Remarks to the Author):

The authors have responded appropriately to my comments.
Lars Klareskog

A point-to-point response to reviewers' comments

Reviewer #2 (Remarks to the Author):

The amount of data and the number of associations is very substantial. Supplemental table 3 is helpful at summarizing the data and providing potential mechanistic insights.

Response: Thank you very much for your nice comments.

I reread the discussion hoping for insights into mechanistic differences between ACPA+/- RA, but was not rewarded. Much repetition of the data. When discussing the different histologic subsets described by others, inferences were made. However, characterization of the histologic data of the samples used in this study was not made. Figure 4e is just an example.

Response: Thank you for the helpful suggestion. We have summarized the major mechanistic differences between the two subsets of RA in the last paragraph of Discussion in our revised manuscript.

“In ACPA- RA ST, inflammatory myeloid cells including macrophage M1 Mk (IL1B) and MMP3 secreting DC, Mk subsets contributed to synovial pathogenesis while in ACPA+ RA ST, lymphoid cells (B cells and T cells) are major players. Therefore, they might respond to different therapeutic strategies.”

Despite we inferred that ACPA- RA patients are more likely clustered into the diffuse-myeloid or pauci-immune category since we found decreased plasma cell genes expression, absence of exhausted T markers and decreased cytotoxic T/NK cells in ST of ACPA- RA, we did not perform histologic staining of CD20 and CD68 in our present study. Due to the limited tissue size we could obtain, most of the ST samples of ACPA- RA have been used for scRNA-sequencing. Further efforts in our future study would be made to confirm our inferences as you have kindly suggested.

I believe that a sentence or two should be included in the introduction to better explain the circumstances surrounding the acquisition of the tissue should be provided. It would be helpful for the reader to understand.

Response: Thank you for the helpful comment. We have added a sentence to explain the circumstances surrounding the acquisition of the tissue in the first paragraph of Results Section.

Reviewer #3 (Remarks to the Author):

The authors have responded appropriately to my comments.

Lars Klareskog

Response: We thank Dr. Lars Klareskog for all the comments on our work.